# SoundCTM: Unifying Score-based and Consistency Models for Full-band Text-to-Sound Generation

**Koichi Saito**[1]    **Dongjun Kim**[2]    **Takashi Shibuya**[1]    **Chieh-Hsin Lai**[1]
**Zhi Zhong**[3]    **Yuhta Takida**[1]    **Yuki Mitsufuji**[1,3]
[1]Sony AI    [2]Stanford University    [3]Sony Group Corporation
Koichi.Saito@sony.com

## Abstract

Sound content creation, essential for multimedia works such as video games and films, often involves extensive trial-and-error, enabling creators to semantically reflect their artistic ideas and inspirations, which evolve throughout the creation process, into the sound. Recent high-quality diffusion-based Text-to-Sound (T2S) generative models provide valuable tools for creators. However, these models often suffer from slow inference speeds, imposing an undesirable burden that hinders the trial-and-error process. While existing T2S distillation models address this limitation through 1-step generation, the sample quality of 1-step generation remains insufficient for production use. Additionally, while multi-step sampling in those distillation models improves sample quality itself, the semantic content changes due to their lack of deterministic sampling capabilities. Thus, developing a T2S generative model that allows creators to efficiently conduct trial-and-error while producing high-quality sound remains a key challenge. To address these issues, we introduce Sound Consistency Trajectory Models (SoundCTM), which allow flexible transitions between high-quality 1-step sound generation and superior sound quality through multi-step deterministic sampling. This allows creators to efficiently conduct trial-and-error with 1-step generation to semantically align samples with their intention, and subsequently refine sample quality with preserving semantic content through deterministic multi-step sampling. To develop Sound-CTM, we reframe the CTM training framework, originally proposed in computer vision, and introduce a novel feature distance using the teacher network for a distillation loss. Additionally, while distilling classifier-free guided trajectories, we introduce a $\nu$-sampling, a new algorithm that offers another source of quality improvement. For the $\nu$-sampling, we simultaneously train both conditional and unconditional student models. For production-level generation, we scale up our model to 1B trainable parameters, making SoundCTM-DiT-1B the first large-scale distillation model in the sound community to achieve both promising high-quality 1-step and multi-step full-band (44.1kHz) generation. Audio samples are available at https://anonymus-soundctm.github.io/soundctm_iclr/.

## 1 Introduction

Sound contents play a pivotal role in multimedia experiences, such as video games, music, and films. In a video game development, for example, sound creators and foley artists meticulously craft sound contents like footsteps, dragon roars, and ambient bird chirps to enhance the immersive quality of gameplay. Recently, Text-to-Sound (T2S) generative models based on Latent Diffusion Models (LDMs) (Rombach et al., 2022), such as Stable Audio (Evans et al., 2024a;c) and AudioLDM2-48kHz (Liu et al., 2024b), demonstrating full-band sound generation in either 44.1 kHz or 48 kHz formats, which is required for real-world applications, have emerged as appealing tools for streamlining the sound production process. Despite their potential, these DMs (Sohl-Dickstein et al., 2015; Song et al., 2021b; Karras et al., 2022) are computationally intensive, making it challenging to swiftly modify and refine sound content to align with creators' continuously evolving artistic inspiration,

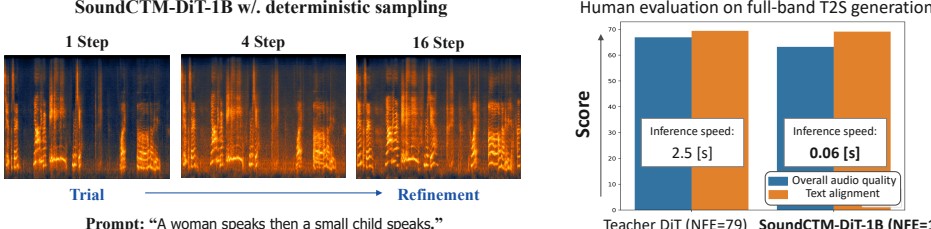

Figure 1: SoundCTM-DiT-1B is first model that achieves high-quality 1-step and higher-quality multi-step full-band T2S generation while preserving semantic content through deterministic sampling, enabling creators to efficiently carry out the trial-and-refinement creation process within a single model.

which is influenced by various triggers, such as the intermediate sounds produced during creation and shifts in their viewpoint toward the desired sound. This paper addresses the issue of slow inference speeds in T2S models and seeks to enhance their editability and practical application.

Sound creation typically involves a significant trial-and-error process, whether through mixing and splicing digital sounds from extensive high-quality sound libraries or recording physical objects. This trial-and-error process, while time-consuming, is crucial for creators to semantically reflect their changing ideas and inspirations throughout the creation, driven by their vision of the experience they want to deliver to consumers, into the sound content.

When applying sound generative models to the sound creation for production use, existing LDM-based full-band T2S models can be leveraged for their high-quality generated sounds. However, due to their slow inference speeds, these models struggle to efficiently accommodate the trial-and-error process, posing an undue burden on creators. While these models suffer from slow inference speeds, recent Consistency Distillation (CD) (Song et al., 2023)-based T2S models, such as ConsistencyTTA (Bai et al., 2023) and AudioLCM (Liu et al., 2024c), offer the potential to accelerate the trial-and-error process with their 1-step generation. However, the sample quality of their 1-step generation remains insufficient for production-level use.

To integrate sound generative models into real-world sound creation, we aim to propose a single model that efficiently facilitates a trial-and-refinement creation process, where creators can first conduct trials with 1-step generation and, once the semantically desired content is obtained, refine the sample quality through multi-step deterministic sampling while preserving the semantic content. It is true that existing CD-based distillation models can improve the sound quality itself through multi-step sampling. However, the refinement phase—where sounds are further refined for production use while preserving their semantic content after the trial phase—remains challenging with those distillation models. This difficulty stems from the fact that deterministic sampling, essential for maintaining semantic content, is not feasible due to their CD's training regime, which only learns anytime-to-zero-time jumps. Therefore, developing a new model that can be applied efficiently to both the trial and refinement phases remains a critical challenge for sound generative models.

To address this challenge, in this paper, we present the ***Sound Consistency Trajectory Model*** (**SoundCTM**), a novel T2S model that enables flexible switching between 1-step high-quality sound generation and higher-quality multi-step generation with deterministic sampling, allowing creators to efficiently perform the trial-and-refinement creation process within a single model (See Figure 1). This is achieved through a training framework that learns anytime-to-anytime jumps (i.e., deterministic mapping) and employs deterministic sampling as proposed in Consistency Trajectory Models (CTMs) (Kim et al., 2024). Building upon the CTM framework, originally proposed in the computer vision field, we address the limitations of its training approach, particularly its heavy reliance on extra pretrained networks to achieve notable generation performance, which are not always accessible in other domains. Specifically, we propose a novel feature distance that uses a teacher network as a feature extractor for distillation loss (see Section 3.1). Furthermore, while distilling classifier-free guided text-conditional trajectories with Classifier-Free Guidance (CFG) (Ho and Salimans, 2022) as a new condition for student models, we introduce $\nu$-sampling, a new sampling algorithm that incorporates the text-conditional and unconditional student models. For $\nu$-sampling, we train both text-conditional and unconditional student models, simultaneously. (see Section 3.2 and Algorithm 1).

In our experiments, SoundCTM shows notable 1-step generation and higher-quality multi-step generation not only in the 16kHz setting but also in the 44.1kHz full-band T2S generation setting, which is the minimum requirement for real-world sound creation. Additionally, by utilizing deterministic sampling, SoundCTM demonstrates the capability to flexibly control the trade-off between inference speed and sample quality while preserving semantic content, thereby enabling the trial-and-refinement process with a single model. We highlight that SoundCTM-DiT-1B, whose teacher model is a DiT-based LDM with 1B trainable parameters, is the first large-scale full-band T2S distillation model capable of achieving both high-quality 1-step generation and higher-quality multi-step generation.

Our contributions are summarized as:

- We introduce SoundCTM, enabling the efficient trial-and-refinement creation process with a single T2S model through high-quality 1-step generation and higher-quality generation with multi-step deterministic sampling.
- To develop SoundCTM, we address the limitations of the CTM framework by proposing a novel feature distance for distillation loss, a strategy for distilling CFG trajectories, and a $\nu$-sampling that combines text-conditional and unconditional student jumps.
- We demonstrate that SoundCTM-DiT-1B is the first large-scale distillation model to achieve notable 1-step and multi-step full-band text-to-sound generation.

## 2 PRELIMINARY

**Diffusion Models**  Let $p_{\text{data}}$ denote the data distribution. In DMs, the data variable $\mathbf{x}_0 \sim p_{\text{data}}$ is generated through a reverse-time stochastic process (Anderson, 1982) defined as $\mathrm{d}\mathbf{x}_t = -2t\nabla \log p_t(\mathbf{x}_t)dt + \sqrt{2t}\,\mathrm{d}\bar{\mathbf{w}}_t$ from time $T$ to 0, where $\bar{\mathbf{w}}_t$ is the standard Wiener process in reverse-time. The marginal density $p_t(\mathbf{x})$ is obtained by encoding $\mathbf{x}_0$ along with a fixed forward diffusion process, $\mathrm{d}\mathbf{x}_t = \sqrt{2t}\,\mathrm{d}\mathbf{w}_t$, initialized by $\mathbf{x}_0$, where $\mathbf{w}_t$ is the standard Wiener process in forward-time. Song et al. (2021b) present the deterministic counterpart of the reverse-time process, called the *Probability Flow* Ordinary Differential Equation (PF ODE), given by

$$\frac{\mathrm{d}\mathbf{x}_t}{\mathrm{d}t} = -t\nabla \log p_t(\mathbf{x}_t) = \frac{\mathbf{x}_t - \mathbb{E}_{p_{t|0}(\mathbf{x}|\mathbf{x}_t)}[\mathbf{x}|\mathbf{x}_t]}{t},$$

where $p_{t|0}(\mathbf{x}|\mathbf{x}_t)$ is the probability distribution of the solution of the reverse-time stochastic process from time $t$ to 0, initiated from $\mathbf{x}_t$. $\mathbb{E}_{p_{t|0}(\mathbf{x}|\mathbf{x}_t)}[\mathbf{x}|\mathbf{x}_t] = \mathbf{x}_t + t\nabla \log p_t(\mathbf{x}_t)$ is a denoiser function[1] (Efron, 2011).

Practically, the denoiser $\mathbb{E}[\mathbf{x}|\mathbf{x}_t]$ is estimated by a neural network $D_\phi$, obtained by minimizing a Denoising Score Matching (DSM) loss (Vincent, 2011; Song et al., 2021b) $\mathbb{E}_{\mathbf{x}_0,t,p_{0|t}(\mathbf{x}|\mathbf{x}_0)}[\|\mathbf{x}_0 - D_\phi(\mathbf{x},t)\|_2^2]$, where $p_{0|t}(\mathbf{x}|\mathbf{x}_0)$ is the transition probability from time 0 to $t$, initiated with $\mathbf{x}_0$. Given the trained denoiser, the empirical PF ODE is given by

$$\frac{\mathrm{d}\mathbf{x}_t}{\mathrm{d}t} = \frac{\mathbf{x}_t - D_\phi(\mathbf{x}_t,t)}{t}. \tag{1}$$

DMs can generate samples by solving the empirical PF ODE, initiated with $\mathbf{x}_T$, which is sampled from a prior distribution $\pi$ approximating $p_T$.

**Text-Conditional Sound Generation with Latent Diffusion Models**  LDM-based T2S models (Liu et al., 2023; 2024b; Ghosal et al., 2023; Evans et al., 2024a;c) generate audio matched to textual descriptions by first obtaining the latent counterpart of the data variable $\mathbf{z}_0$ through the reverse-time process conditioned by text embedding $\mathbf{c}_{\text{text}}$. This latent variable $\mathbf{z}_0$ is then converted to $\mathbf{x}_0$ using a pretrained decoder $\mathcal{D}$. During the training phase, $D_\phi$ is trained by minimizing the DSM loss $\mathbb{E}_{\mathbf{z}_0,t,p_{0|t}(\mathbf{z}|\mathbf{z}_0)}[\|\mathbf{z}_0 - D_\phi(\mathbf{z},t,\mathbf{c}_{\text{text}})\|_2^2]$, where $p_{0|t}(\mathbf{z}|\mathbf{z}_0)$ is the latent counterpart of the transition probability from time 0 to $t$, initiated with $\mathbf{z}_0$. $\mathbf{z}_0$ is given by a pretrained encoder as $\mathbf{z}_0 = \mathcal{E}(\mathbf{x}_0)$. We refer to Appendix A for a review of related work.

---

[1] For simplicity, we omit $p_{t|0}(\mathbf{x}|\mathbf{x}_t)$, a subscript in the expectation of the denoiser, throughout the paper.

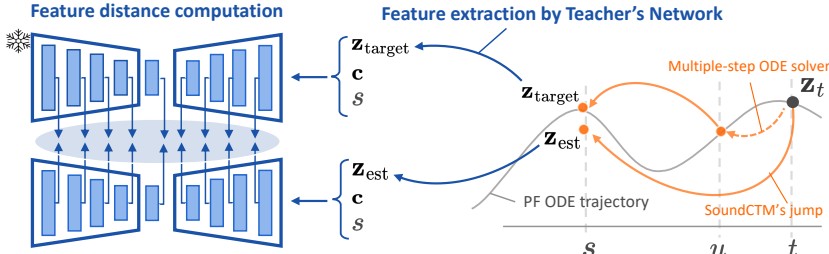

Figure 2: Illustrations of SoundCTM's two predictions $\mathbf{z}_{\text{target}}$ and $\mathbf{z}_{\text{est}}$ at time $s$ with an initial value $\mathbf{z}_t$ and the feature extraction by the teacher's network for the CTM loss shown within the blue ellipse area. All the parameters of the teacher's network are frozen. The conditional embedding $\mathbf{c}$ and time $s$ are also input to the feature extractor. Note that the teacher's network does not need to be the UNet architecture (Ronneberger et al., 2015).

**Consistency Models**  Consistency Models (CMs) (Song et al., 2023) and CD predict anytime-to-zero time long jumps of the PF ODE trajectory. $G(\mathbf{x}_t, t, 0)$ is defined as the solution of the PF ODE from initial time $t$ to final time $0$, and $G$ is estimated by $G_{\boldsymbol{\theta}}$ as the neural jump. To train $G_{\boldsymbol{\theta}}$, two time step 0-predictions are compared: one from a teacher $\phi$ and the other from a student $\boldsymbol{\theta}$ as:

$$G_{\boldsymbol{\theta}}(\mathbf{x}_t, t, 0) \approx G_{\text{sg}(\boldsymbol{\theta})}\big(\texttt{Solver}(\mathbf{x}_t, t, t - \Delta t; \phi), t - \Delta t, 0\big),$$

where $\texttt{Solver}(\mathbf{x}_t, t, t - \Delta t; \phi)$ is the pre-trained PF ODE in Eq. (1) within the interval $[t - \Delta t, t]$, which determines the amount of teacher information to distill, and $\texttt{sg}$ is the exponential moving average (EMA) stop-gradient $\texttt{sg}(\boldsymbol{\theta}) \leftarrow \texttt{stopgrad}(\mu \texttt{sg}(\boldsymbol{\theta}) + (1 - \mu)\boldsymbol{\theta})$. In CMs and CD, only stochastic sampling is possible during multi-step sampling. This is because the model is trained using the anytime-to-zero time jump framework, which requires adding noise to the estimated $\mathbf{x}_0$ to obtain $\mathbf{x}_t$ for multi-step sampling. As a result, the semantic content of generated samples changes depending on the number of sampling steps. This variability hinders the trial-and-refinement process, which is why we do not adopt the CD framework.

**Consistency Trajectory Models**  In contrast to CMs, CTMs predict both infinitesimally small step jump and long step jump of the PF ODE trajectory. $G(\mathbf{x}_t, t, s)$ is defined as the solution of the PF ODE from initial time $t$ to final time $s \leq t$, and $G$ is estimated by $G_{\boldsymbol{\theta}}$ as the neural jump. To train $G_{\boldsymbol{\theta}}$, two $s$-predictions are compared: one from a teacher $\phi$ and the other from a student $\boldsymbol{\theta}$ as:

$$G_{\boldsymbol{\theta}}(\mathbf{x}_t, t, s) \approx G_{\text{sg}(\boldsymbol{\theta})}\big(\texttt{Solver}(\mathbf{x}_t, t, u; \phi), u, s\big), \tag{2}$$

where $\texttt{Solver}(\mathbf{x}_t, t, u; \phi)$ is the pre-trained PF ODE in Eq. (1), a random $u \in [s, t)$ determines the amount of teacher information to distill. To quantify the dissimilarity (CTM loss) between the student prediction $G_{\boldsymbol{\theta}}(\mathbf{x}_t, t, s)$ and the teacher prediction $G_{\text{sg}(\boldsymbol{\theta})}\big(\texttt{Solver}(\mathbf{x}_t, t, u; \phi), u, s\big)$ in Eq. (2), the Learned Perceptual Image Patch Similarity (LPIPS) (Zhang et al., 2018) is used to measure a feature distance $d_{\text{feat.}}$ after transporting both predictions from $s$-time to 0-time as $\mathbf{x}_{\text{est}}(\mathbf{x}_t, t, s) :=$ $G_{\text{sg}(\boldsymbol{\theta})}(G_{\boldsymbol{\theta}}(\mathbf{x}_t, t, s), s, 0)$ and $\mathbf{x}_{\text{target}}(\mathbf{x}_t, t, u, s) := G_{\text{sg}(\boldsymbol{\theta})}(G_{\text{sg}(\boldsymbol{\theta})}\big(\texttt{Solver}(\mathbf{x}_t, t, u; \phi), u, s\big), s, 0)$. Summarizing, the CTM loss is defined as

$$\mathcal{L}_{\text{CTM}}(\boldsymbol{\theta}; \phi) := \mathbb{E}_{t \in [0, T]} \mathbb{E}_{s \in [0, t]} \mathbb{E}_{u \in [s, t)} \mathbb{E}_{\mathbf{x}_0} \mathbb{E}_{\mathbf{x}_t | \mathbf{x}_0} \Big[ d_{\text{feat.}}\big(\mathbf{x}_{\text{target}}(\mathbf{x}_t, t, u, s), \mathbf{x}_{\text{est}}(\mathbf{x}_t, t, s)\big) \Big]. \tag{3}$$

With this anytime-to-anytime jump training framework, both stochastic and deterministic sampling are possible in CTMs, enabling the trial-and-refinement process for sound creation.

## 3  SOUNDCTM

To address the challenges of achieving fast, flexible, and high-quality T2S generation, we introduce SoundCTM by reframing the CTM's training framework. Consistent with the CTM, we use the same distillation loss. Since this paper primarily illustrates the method using LDM-based T2S models as the teacher model, the student model is trained to estimate the neural jump $G_{\boldsymbol{\theta}}$ as:

$$G_{\boldsymbol{\theta}}(\mathbf{z}_t, \mathbf{c}_{\text{text}}, t, s) \approx G_{\text{sg}(\boldsymbol{\theta})}\big(\texttt{Solver}(\mathbf{z}_t, \mathbf{c}_{\text{text}}, t, u; \phi), \mathbf{c}_{\text{text}}, u, s\big), \tag{4}$$

where $\mathbf{z}_t$ is the latent counterpart of $\mathbf{x}_t$, and $\mathtt{Solver}(\mathbf{z}_t, \mathbf{c}_{\text{text}}, t, u; \boldsymbol{\phi})$ is the numerical solver of the pre-trained text-conditional PF ODE by following Eq. (2). To quantify the dissimilarity between $G_{\boldsymbol{\theta}}$ and $G_{\mathtt{sg}(\boldsymbol{\theta})}$, we propose a new feature distance in Section 3.1. We refer to Appendix A for a review of related work about our proposed feature distance.

## 3.1 TEACHER'S NETWORK AS FEATURE EXTRACTOR FOR CTM LOSS

We propose a new training framework that **utilizes the teacher's network as a feature extractor** and measures the feature distance $d_{\text{teacher}}$ for the CTM loss between $G_{\boldsymbol{\theta}}$ and $G_{\mathtt{sg}(\boldsymbol{\theta})}$ in Eq. (4), as illustrated in Figure 2. Utilizing $d_{\text{teacher}}$ offers two benefits:

- Using $d_{\text{teacher}}$ yields better performance compared with using the $l_2$ distance in the $\mathbf{z}$ domain computed at either 0-time or $s$-time, as demonstrated in Table 1.
- The teacher network can naturally be utilized even in situations where external off-the-shelf pretrained networks for computing a feature distance are inaccessible, since SoundCTM is a distillation model that assumes the availability of a teacher diffusion model. We discuss the potential use of external off-the-shelf pretrained networks in Appendix B.

We define $d_{\text{teacher}}$ between two predictions $\mathbf{z}_{\text{target}}$ and $\mathbf{z}_{\text{est}}$ as follows:

$$d_{\text{teacher}}(\mathbf{z}_{\text{target}}, \mathbf{z}_{\text{est}}, \mathbf{c}, t) = \sum_{m=1}^{M} \|\text{TN}_{\boldsymbol{\phi},m}(\mathbf{z}_{\text{target}}, \mathbf{c}, t) - \text{TN}_{\boldsymbol{\phi},m}(\mathbf{z}_{\text{est}}, \mathbf{c}, t)\|_2^2, \quad (5)$$

where $\text{TN}_{\boldsymbol{\phi},m}(\cdot, \mathbf{c}, t)$ denotes the channel-wise normalized [2] output feature of the $m$-th layer of the pretrained teacher's network, conditioned by time $t$ and embedding $\mathbf{c}$. This approach is feasible since noisy latents are input to the teacher's network during teacher's training.

## 3.2 CFG HANDLING FOR SOUNDCTM

CFG plays a pivotal role in T2S generation, as well as in other modalities (Ho and Salimans, 2022), for generating high-quality samples. To leverage this advantage of CFG, we propose distilling the classifier-free guided PF ODE trajectory scaled by $\omega$ uniformly sampled from the range $[\omega_{\min}, \omega_{\max}]$ during training, and using $\omega$ as a new condition in the student network defined as:

$$G_{\boldsymbol{\theta}}(\mathbf{z}_t, \mathbf{c}_{\text{text}}, \omega, t, s) = \frac{s}{t}\mathbf{z}_t + \left(1 - \frac{s}{t}\right)g_{\boldsymbol{\theta}}(\mathbf{z}_t, \mathbf{c}_{\text{text}}, \omega, t, s), \quad (6)$$

where $g_{\boldsymbol{\theta}}$ is a neural output. To summarize Eqs.(4)–(6), the distillation loss for SoundCTM is formulated as:

$$\mathcal{L}_{\text{CTM}}^{\text{Sound}}(\boldsymbol{\theta}; \boldsymbol{\phi}) := \mathbb{E}_{t,s,u,\omega,\mathbf{z}_t}\left[d_{\text{teacher}}\big(\mathbf{z}_{\text{target}}(\mathbf{z}_t, \mathbf{c}_{\text{text}}, \omega, t, u, s), \mathbf{z}_{\text{est}}(\mathbf{z}_t, \mathbf{c}_{\text{text}}, \omega, t, s), \mathbf{c}_{\text{text}}, s\big)\right], \quad (7)$$

where

$$\mathbf{z}_{\text{target}}(\mathbf{z}_t, \mathbf{c}_{\text{text}}, \omega, t, u, s) := G_{\mathtt{sg}(\boldsymbol{\theta})}\big(\mathtt{Solver}(\mathbf{z}_t, \mathbf{c}_{\text{text}}, \omega, t, u; \boldsymbol{\phi}), \mathbf{c}_{\text{text}}, \omega, u, s\big),$$

$$\mathbf{z}_{\text{est}}(\mathbf{z}_t, \mathbf{c}_{\text{text}}, \omega, t, s) := G_{\boldsymbol{\theta}}(\mathbf{z}_t, \mathbf{c}_{\text{text}}, \omega, t, s),$$

$\mathtt{Solver}(\mathbf{z}_t, \mathbf{c}_{\text{text}}, \omega, t, u; \boldsymbol{\phi}) := \omega\mathtt{Solver}(\mathbf{z}_t, \mathbf{c}_{\text{text}}, t, u; \boldsymbol{\phi}) + (1-\omega)\mathtt{Solver}(\mathbf{z}_t, \varnothing, t, u; \boldsymbol{\phi})$, $\varnothing$ is an unconditional embedding, $u \in [s, t)$, and $\omega \sim U[\omega_{\min}, \omega_{\max}]$, respectively.

To achieve better generation performance, Kim et al. (2024) use the two auxiliary losses, the DSM loss and an adversarial loss (GAN loss) (Goodfellow et al., 2014). For SoundCTM, we only use the DSM loss, defined as:

$$\mathcal{L}_{\text{DSM}}^{\text{Sound}}(\boldsymbol{\theta}) = \mathbb{E}_{t,\mathbf{z}_0,\omega,\mathbf{z}_t|\mathbf{z}_0}[\|\mathbf{z}_0 - g_{\boldsymbol{\theta}}(\mathbf{z}_t, \mathbf{c}_{\text{text}}, \omega, t, t)\|_2^2]. \quad (8)$$

Note that the DSM loss serves to improve the accuracy of approximating the small jumps during the training. We discuss the reason why we eliminate the GAN loss in Section C. Summing Eqs. (7) and (8), SoundCTM is trained with the following objective:

$$\mathcal{L}(\boldsymbol{\theta}) := \mathcal{L}_{\text{CTM}}^{\text{Sound}}(\boldsymbol{\theta}; \boldsymbol{\phi}) + \lambda_{\text{DSM}}\mathcal{L}_{\text{DSM}}^{\text{Sound}}(\boldsymbol{\theta}), \quad (9)$$

where $\lambda_{\text{DSM}}$ is a scaling weight for $\mathcal{L}_{\text{DSM}}^{\text{Sound}}$. We summarize SoundCTM's training in Appendix D.

---

[2]The idea of this normalization process is borrowed from LPIPS (Zhang et al., 2018, Sec. 3).

### 3.3 SAMPLING SCHEME OF SOUNDCTM

Inspired by the performance improvement by using CFG in DMs, we introduce $\nu$-sampling in SoundCTM's sampling, which incorporates the text-conditional and unconditional student models, given by:

$$\mathbf{z}_{s|t} = G_{\boldsymbol{\theta}}(\mathbf{z}_t, \varnothing, \omega, t, s) + \nu(G_{\boldsymbol{\theta}}(\mathbf{z}_t, \mathbf{c}_{\text{text}}, \omega, t, s) - G_{\boldsymbol{\theta}}(\mathbf{z}_t, \varnothing, \omega, t, s)). \quad (10)$$

Algorithm 1 summarizes SoundCTM's sampling, with the sampling timesteps denoted as $T = t_0 > \cdots > t_N = 0$. Note that in contrast to the previous sampling methods for the diffusion-based distillation models (Bai et al., 2023; Liu et al., 2024c; Kim et al., 2024), we also use $G_{\boldsymbol{\theta}}(\mathbf{z}_t, \varnothing, \omega, t, s)$ as highlighted in blue Algorithm 1. We train both $G_{\boldsymbol{\theta}}(\mathbf{z}_t, \mathbf{c}_{\text{text}}, \omega, t, s)$ and $G_{\boldsymbol{\theta}}(\mathbf{z}_t, \varnothing, \omega, t, s)$ simultaneously as shown in Algorithm 2.

---

**Algorithm 1** SoundCTM's Inference

**Require:** Hyperparameter $\nu$, text condition $\mathbf{c}_{\text{text}}$, CFG scale $\omega$, hyperparameter of CTM's $\gamma$-sampling $\gamma$

1: Sample $\mathbf{z}_{t_0}$ from prior distribution
2: **for** $n = 0$ to $N - 1$ **do**
3:     $\tilde{t}_{n+1} \leftarrow \sqrt{1 - \gamma^2} t_{n+1}$
4:     $\mathbf{z}_{\tilde{t}_{n+1}} \leftarrow \nu G_{\boldsymbol{\theta}}(\mathbf{z}_{t_n}, \mathbf{c}_{\text{text}}, \omega, t_n, \tilde{t}_{n+1})$
5:         $+(1-\nu)G_{\boldsymbol{\theta}}(\mathbf{z}_{t_n}, \varnothing, \omega, t_n, \tilde{t}_{n+1})$
6:     $\mathbf{z}_{t_{n+1}} \leftarrow \mathbf{z}_{\tilde{t}_{n+1}} + \gamma t_{n+1}\boldsymbol{\epsilon}$
7: **end for**
8: **Return** $\mathbf{z}_{t_N}$

---

## 4 EXPERIMENTS

### 4.1 T2S GENERATION ON 16KHZ

As an initial step, in this section, we conduct experiments on 16 kHz audio signals by following most of the existing T2S generative models (Ghosal et al., 2023; Liu et al., 2023; 2024b; Bai et al., 2023; Liu et al., 2024c). We evaluate SoundCTM on the AudioCaps dataset (Kim et al., 2019), which contains $47,289$ pairs of 10-second audio samples and human-written text descriptions for the training set and $957$ samples for the testset. All audio samples are downsampled to 16 kHz. We adopt TANGO (Ghosal et al., 2023) as the teacher model trained with EDM's variance exploding formulation (Karras et al., 2022). We use deterministic sampling ($\gamma = 0$ in Algorithm 1) and evaluate the model performance with student EMA rate $\mu = 0.999$. Experimental details are described in Appendix D.1.

**Evaluation Metrics** We use four objective metrics: the Frechet Audio Distance ($\text{FAD}_{\text{vgg}}$) (Kilgour et al., 2019) on VGGish, the Kullback-Leibler divergence ($\text{KL}_{\text{passt}}$) on PaSST (Koutini et al., 2022), a state-of-the-art audio classification model, the Inception Score ($\text{IS}_{\text{passt}}$) (Salimans et al., 2016) on PaSST, and the CLAP score[3] (Wu* et al., 2023).

**Effectiveness of Utilizing Teacher's Network as Feature Extractor** We first evaluate the efficacy of utilizing the teacher's network as a feature extractor in Eq (7), which we newly proposed in Section 3.1, by comparing the following cases: Using 1) the $l_2$ at 0-time step, 2) the $l_2$ at $s$-time step, and 3) the $d_{\text{teacher}}$ at $s$-time step. We use the entire layers of the teacher's network for computing $d_{\text{teacher}}$.

As shown in Table 1, using $d_{\text{teacher}}$ demonstrates better performance across all the metrics against all the other cases. This result indicates that using $d_{\text{teacher}}$ enables the student model to distill the trajectory more accurately than using $l_2$. In the subsequent experiments, unless otherwise noted, SoundCTM refers to the model trained with using $d_{\text{teacher}}$.

Table 1: Effectiveness of using proposed $d_{\text{teacher}}$ against $l_2$ on $z_0$ space evaluated on AudioCaps testset. $\omega = 3.5$ and $\nu = 1.0$ are used for inference.

| Model | # of steps | $\text{FAD}_{\text{vgg}} \downarrow$ | $\text{KL}_{\text{passt}} \downarrow$ | $\text{IS}_{\text{passt}} \uparrow$ | CLAP $\uparrow$ |
|---|---|---|---|---|---|
| Teacher diffusion model (TANGO-EDM w/. Heun solver) | 40 | 1.71 | 1.28 | 8.11 | 0.46 |
| **Student Models** | | | | | |
| SoundCTM w/. $l_2$ (0-time step) | 1 | 2.43 | 1.28 | 6.87 | 0.42 |
| SoundCTM w/. $l_2$ ($s$-time step) | 1 | 2.45 | 1.28 | 6.83 | 0.42 |
| **SoundCTM w/. $d_{\text{teacher}}$** | 1 | **2.17** | **1.27** | **7.18** | **0.43** |

**Preserving Semantic Content within Multi-step Sampling** One of our goals is to develop a model capable of achieving high-quality 1-step generation and higher-quality multi-step generation

---

[3]We use the "630k-audioset-best.pt" checkpoint from https://github.com/LAION-AI/CLAP

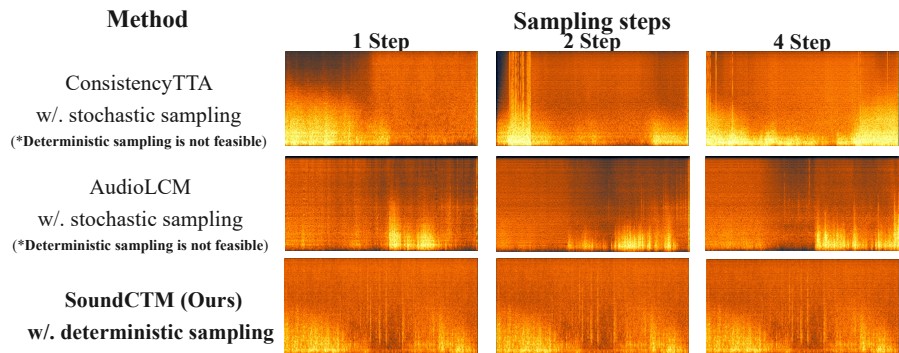

Prompt: "Thunder claps, and hard rain falls and splashes on surfaces."

Figure 3: Visualization of spectrograms of generated samples using 1-step, 2-step, and 4-step generation with ConsistencyTTA, AudioLCM, and SoundCTM.

Table 2: Performance comparisons on AudioCaps testset at 16 kHz. **Bold scores indicate the best results under 1-step generation. Underlined scores indicate the best results under multi-step generation of Distillation Models.** † denotes the results tested by us using the open-sourced checkpoints, as not all metrics are provided in each paper.

| Model | # of sampling steps | $\omega$ | $\nu$ | $FAD_{vgg} \downarrow$ | $KL_{passt} \downarrow$ | $IS_{passt} \uparrow$ | $CLAP \uparrow$ |
|---|---|---|---|---|---|---|---|
| **Diffusion Models** | | | | | | | |
| AudioLDM 2-AC-Large (Liu et al., 2024b) | 200 | 3.5 | - | 1.42 | 0.98 | - | - |
| TANGO (Ghosal et al., 2023) | 200 | 3.0 | - | 1.64 | 1.31† | 6.35† | 0.44† |
| Teacher of ConsistencyTTA (Bai et al., 2023) | 200 | 3.0 | - | 1.91 | - | - | - |
| Teacher of AudioLCM (Liu et al., 2024c) | 100 | 5.0 | - | 1.56 | - | - | - |
| Our Teacher model (TANGO w/. Heun Solver) | 40 | 3.5 | - | 1.71 | 1.28 | 8.11 | 0.46 |
| **Distillation Models** | | | | | | | |
| ConsistencyTTA (Bai et al., 2023) | 1 | 4.0 | - | 2.41 | 1.31† | **7.84**† | 0.42† |
| | 2 | 4.0 | - | 2.48† | 1.27† | 7.88† | 0.41† |
| | 4 | 4.0 | - | 3.01† | 1.31† | 8.05† | 0.42† |
| AudioLCM (Liu et al., 2024c) | 1 | 5.0 | - | 4.04† | 1.75† | 6.52† | 0.33† |
| | 2 | 5.0 | - | 1.67 | 1.49† | 8.24† | 0.41† |
| | 4 | 5.0 | - | 1.42† | 1.40 | 8.78† | 0.42† |
| **SoundCTM (Ours)** | 1 | 3.5 | 1.0 | **2.08** | **1.26** | 7.13 | **0.43** |
| | 2 | 3.5 | 1.0 | 1.90 | 1.24 | 7.26 | 0.45 |
| | 4 | 3.5 | 1.0 | 1.72 | 1.22 | 7.37 | 0.45 |
| | 8 | 3.0 | 1.5 | 1.45 | 1.20 | 7.98 | 0.46 |
| | 16 | 3.0 | 2.0 | 1.38 | 1.19 | 8.24 | 0.46 |

while preserving semantic content through deterministic sampling. To demonstrate this capability, we visualize the spectrograms of generated samples from the same initial noise and input text prompt across different numbers of sampling steps with SoundCTM's deterministic sampling and other T2S distillation models (ConsistencyTTA and AudioLCM) in Figure 3.

We emphasize that, by its anytime-to-anytime jump training framework, SoundCTM can preserve generated content using deterministic sampling ($\gamma = 0$) [4], even when varying the number of sampling steps (see additional examples in Figure 6). This cannot be achieved with ConsistencyTTA or AudioLCM due to their lack of deterministic sampling capability. In terms of the sample quality, as shown in Table 2, which we discuss later, SoundCTM shows clear trade-offs between the performance improvements and the number of sampling steps. These SoundCTM's characteristics allow sound creators to efficiently proceed with the trial-and-refinement process with a single model.

**Performance Comparison with Other T2S Models**  We compare SoundCTM with other T2S models under both 1-step and multi-step generation. We employ ConsistencyTTA and AudioLCM as our baseline models. We also include AudioLDM2-AC-Large (Liu et al., 2024b), TANGO (Ghosal

---

[4]Note that SoundCTM also supports stochastic sampling by adjusting $\gamma$.

Table 3: Performance comparisons on AudioCaps at full-band setting. Bold scores indicate best results.

| Model | channels/sr | # of sampling steps | $\omega$ | $\nu$ | $FD_{openl3}\downarrow$ | $KL_{passt}\downarrow$ | $CLAP\uparrow$ | Inference speed [sec.] $\downarrow$ |
|---|---|---|---|---|---|---|---|---|
| **Diffusion Models** | | | | | | | | |
| Stable Audio Open (Evans et al., 2024c) | 2/44.1kHz | 100 | 7.0 | - | 78.24 | 2.14 | 0.29 | - |
| AudioLDM2-48kHz (Liu et al., 2024b) | 1/48kHz | 200 | 3.5 | - | 101.11 | 2.04 | 0.37 | - |
| Our teacher model w/. Heun Solver | 1/44.1kHz | 40 | 5.0 | - | 72.34 | 1.74 | 0.36 | 2.50 |
| **Distillation Models** | | | | | | | | |
| **SoundCTM-DiT-1B** | 1/44.1kHz | 1 | 5.0 | 1.0 | 84.07 | 1.74 | 0.36 | **0.06** |
| | | 2 | 5.0 | 1.0 | 79.96 | 1.61 | 0.38 | 0.10 |
| | | 4 | 5.0 | 1.0 | 83.14 | 1.57 | 0.37 | 0.18 |
| | | 8 | 5.0 | 5.0 | 79.05 | 1.29 | 0.40 | 0.35 |
| | | 16 | 5.0 | 5.0 | **72.24** | **1.27** | **0.42** | 0.71 |

et al., 2023) in the evaluation for completeness. Table 2 presents the quantitative results. All the LDM-based models use DDIM sampling (Song et al., 2021a) except for our teacher model.

Under the 1-step generation setting, SoundCTM shows the best performance in all the evaluation metrics except for $IS_{passt}$. Note that, considering its teacher models' performance, the results of 1-step generation of AudioLCM is limited (See also (Liu et al., 2024c, Fig.2 (b)).). Under the multi-step case, AudioLCM and SoundCTM show clear trade-offs between the performance improvements and the number of sampling steps. On the other hand, ConsistencyTTA dose not show such trade-offs, implying that ConsistencyTTA might suffer from accumulated errors during multi-step stochastic sampling. These results indicate that SoundCTM is the first T2S distillation model to successfully achieve both promising 1-step and multi-step generation.

## 4.2 LARGE-SCALE FULL-BAND T2S GENERATION

Inspired by the success of recent large-scale (e.g., 1B scale) LDM-based T2S generative models (Evans et al., 2024a;b;c; Liu et al., 2024b) in achieving full-band T2S generation suitable for real-world applications, for production-level generation, we evaluate the scalability of SoundCTM to a 1B-scale model and its extension to full-band T2S generation tasks. We conduct our evaluation on the AudioCaps dataset, with all audio samples resampled to 44.1 kHz. Following Evans et al. (2024a), we use all 4,875 text captions from the AudioCaps testset, which contains 5 captions per audio clip.

In contrast to the UNet-based model used in Section 4.1, we adopt the diffusion-transformer (DiT) (Peebles and Xie, 2023), which contains 1.0B trainable parameters, as the teacher and student model. The teacher model is trained using EDM's variance exploding formulation. To distinguish the UNet-based student model, we, hereafter, refer to the DiT-based student model as SoundCTM-DiT-1B. We use deterministic sampling and evaluate the model performance with student EMA rate $\mu = 0.96$. We use the entire DiT blocks of the teacher's network for computing $d_{teacher}$. Experimental details are described in Appendix D.2. It is worth noting that in the field of sound generation, SoundCTM-DiT-1B represents the first attempt to distill a LDM-based large-scale full-band T2S generative model.

**Evaluation Metrics** Following the objective evaluation protocol for the full-band T2S generation in Stable Audio series (Evans et al., 2024a;b;c), we compute three objective metrics: the $FD_{openl3}$ (Cramer et al., 2019), $KL_{passt}$, and the CLAP score by using `stable-audio-metrics`[5].

**Quantitative Evaluation** Since SoundCTM is the first attempt to perform a few step full-band T2S generation using a distillation model, we only employ diffusion models such as AudioLDM2-48kHz (Liu et al., 2024b), Stable Audio Open (Evans et al., 2024c) [6], and our teacher model as baselines. AudioLDM2-48kHz, Stable Audio Open, and our teacher model use DDIM sampling (Song et al., 2021a), DPM-Solver++ (Lu et al., 2023), and our 2nd-order Heun solver, respectively.

---

[5] https://github.com/Stability-AI/stable-audio-metrics

[6] Note that the results of Stable Audio Open might not comparable to the other results since AudioCaps is not used for its training, as mentioned in Evans et al. (2024c, Table 2).

Table 4: Subjective evaluation on AudioCaps testset on full-band T2S generation setting. We report overall audio quality (OVAL) and text alignment (REL) with 95% Confidence Interval.

| Method | # of steps | $\omega$ | $\nu$ | OVAL ↑ | REL ↑ |
|---|---|---|---|---|---|
| Ground-truth | - | - | - | $79.6 \pm 1.29$ | $83.2 \pm 1.13$ |
| Stable Audio Open (Evans et al., 2024c) w/. DPM-Solver++ | 100 | 7.0 | - | $54.0 \pm 1.58$ | $47.1 \pm 1.65$ |
| AudioLDM2-48kHz (Liu et al., 2024b) w/. DDIM | 200 | 3.5 | - | $65.1 \pm 1.45$ | $69.9 \pm 1.42$ |
| Our teacher model w/. Heun Solver | 40 | 5.0 | - | $66.9 \pm 1.41$ | $69.4 \pm 1.48$ |
| **SoundCTM-DiT-1B** | 1 | 5.0 | 1.0 | $63.2 \pm 1.46$ | $69.1 \pm 1.34$ |
| | 16 | 5.0 | 5.0 | $\mathbf{72.0 \pm 1.30}$ | $\mathbf{77.0 \pm 1.22}$ |

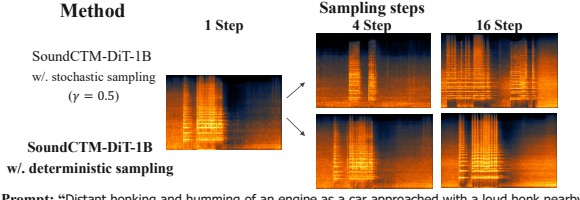

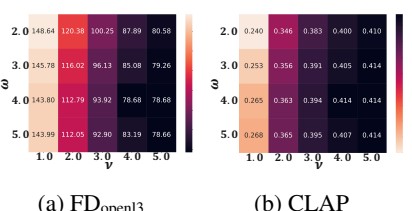

Figure 4: Visualization of spectrograms of generated samples by SoundCTM-DiT-1B using 1-step, 4-step, and 16-step generation with stochastic ($\gamma = 0.5$) and deterministic ($\gamma = 0$) sampling.

(a) FD$_{openl3}$  (b) CLAP

Figure 5: Influence of $\nu$ on SoundCTM-DiT-1B with 8-step sampling. Darker colors indicate better scores.

Table 3 presents the quantitative results. SoundCTM-DiT-1B demonstrates promising performance in both 1-step and multi-step generation, with clear trade-offs between the number of sampling steps and the sample quality. These results indicate that the SoundCTM framework is scalable to the full-band T2S generation setting. We also measured the inference time of SoundCTM-DiT-1B and its teacher diffusion model on a single NVIDIA H100. As shown in Table 3, SoundCTM demonstrates faster generation compared to the teacher model, enabling creators to efficiently conduct the trial-and-refinement creation process with full-band setting.

**Subjective Evaluation** For further evaluation of SoundCTM-DiT-1B, we conduct a listening test. The baselines used for comparison included Stable Audio Open, AudioLDM2-48kHz, our teacher model, SoundCTM with 1-step generation, SoundCTM with 16-step generation, and ground-truth samples. We asked 17 human evaluators to assess the generated audio samples based on two criteria: overall audio quality (OVAL) and relevance to the text prompts (REL). Each participant was presented with 10 samples per model for each metric and asked to rate them on a scale from 1 to 100. The text prompts for the generated samples were randomly selected from the AudioCaps testset. Considering the difficulty of comparing generated samples in a T2S generation task with the single stimulus testing approach, following Evans et al. (2024a), we employ MUSHRA method (Union, 2003) on the webMUSHRA platform (Schoeffler and et al., 2018) as the evaluation protocol (See also Figure 9).

As demonstrated in Table 4, SoundCTM-DiT-1B's 1-step generation shows comparable scores to other methods across both metrics. Moreover, comparing the 1-step and 16-step generation results, the 16-step generation demonstrates higher scores. These results, along with the objective evaluation results in Table 3, indicate that SoundCTM-DiT-1B is capable of flexibly switching between high-quality 1-step and higher-quality multi-step sound generation, even in a full-band setting.

**Stochastic and Deterministic Sampling on SoundCTM-DiT-1B** Once again, one of our goals is to develop a model capable of achieving high-quality 1-step generation and higher-quality multi-step generation while preserving semantic content through deterministic sampling. To demonstrate this capability in SoundCTM-DiT-1B, we visualize the spectrograms of generated samples from the same initial noise and input text prompt across varying numbers of sampling steps with both deterministic and stochastic sampling. As shown in Figure 4 (See more examples in Figure 7 and Figure 8), SoundCTM-DiT-1B preserves semantic content with deterministic sampling while allowing variations through stochastic sampling.

**Influence of $\nu$-sampling**   We examine the influence of both $\omega$ and $\nu$ on multi-step sampling, particularly 8-step generation, for SoundCTM-DiT-1B. In this experiment, we use a single caption per audio clip (957 samples) and compute the $\text{FD}_{\text{openl3}}$ and the CLAP score. As shown in Figure 5, regardless of the value of $\omega$, increasing the value of $\nu$ improves the sample quality. This indicates that the hyperparameter $\nu$ serves a role similar to the CFG scale.

### 4.3   Potential Application: Guidance-based Controllable generation

For further exploration of SoundCTM's application to sound creation, we investigate its potential downstream task and conduct experiments on sound intensity control task (Novack et al., 2024b;a; Wu et al., 2024).

**Loss-based Guidance for SoundCTM**   In DMs, a loss-based guidance method is one of the major method for training-free controllable generation with using pretrained DMs (Yu et al., 2023; Levy et al., 2023). The loss-based guidance method in DMs involves an additional update $\mathbf{z}_{t-1} = \mathbf{z}_{t-1} - \rho_t \nabla_{\mathbf{z}_t} \mathcal{L}(f(\hat{\mathbf{x}}_0), \mathbf{y}_{\text{condition}})$ during sampling, where $f(\cdot)$ is a differentiable feature extractor that converts $\hat{\mathbf{x}}_0$ into the same space as the target condition $\mathbf{y}_{\text{condition}}$, $\hat{\mathbf{x}}_0 = \mathcal{D}(\hat{\mathbf{z}}_0(\mathbf{z}_t))$, $\hat{\mathbf{z}}_0(\mathbf{z}_t)$ is a clean estimate derived from $\mathbf{z}_t$ using Tweedie's formula (Efron, 2011), and $\rho_t$ is a learning rate.

**Sound Intensity Control**   To validate the proposed loss-based guidance framework with Sound-CTM, we conduct sound intensity control (Novack et al., 2024b; Wu et al., 2024; Novack et al., 2024a). This task adjusts the dynamics of the generated sound to match a given target volume line or curve. We follow the experimental protocol from DITTO (Novack et al., 2024b) to control the decibel (dB) volume line or curve of the generated samples. We use 200 audio-text pairs from the AudioCaps testset for each of the 6 different types of $\mathbf{y}_{\text{condition}}$ (See Figure 10 (a) and Figure 11 (a)). We use the pretrained UNet-based SoundCTM obtained from Section 4.1. We evaluate our framework using the mean squared error (MSE) between the target and obtained $\mathbf{y}_{\text{condition}}$, the $\text{FAD}_{\text{vgg}}$, and the CLAP score. Experimental details are provided in Appendix D.3.

We employ DITTO-2 (Novack et al., 2024a) and default T2S generation as baselines. DITTO-2 is a trainig-free controllable generation method using pretrained distillation models proposed in the music generation field by optimizing an initial noise latent $\mathbf{z}_T$ given the target loss. Since DITTO-2 is not open-sourced, we apply its $\mathbf{z}_T$-optimization to SoundCTM.

From Table 5 and the intensity curves shown in Figures 10 to 11, our method successfully controls the sound intensities. These results show the potential capability of the loss-based guidance of SoundCTM. Be aware that the objective of this experiment is to show the potential application of SoundCTM rather than to propose a new method that outperforms other methods for training-free controllable generation.

Table 5: Quantitative results of sound intensity control on SoundCTM. Number of sampling step is 16 for all methods.

| Methods | $\gamma$ | MSE↓ | $\text{FAD}_{\text{vgg}}$ ↓ | CLAP ↑ |
|---|---|---|---|---|
| Default T2S Generation | 0 | 231.9 | 2.08 | 0.47 |
| DITTO-2 (Novack et al., 2024a) | 0 | 60.2 | 3.65 | 0.38 |
| | 0.2 | 50.4 | 3.71 | **0.41** |
| **Our loss-based guidance** | 0 | **18.5** | **3.04** | **0.41** |

## 5   Conclusion

SoundCTM addresses the challenges of current T2S generative models, which make it difficult for creators to efficiently conduct the trial-and-error to semantically reflect their artistic inspirations into the sound and generate high-quality sound within a single model. To develop SoundCTM, we propose a novel feature distance for distillation loss, a strategy for distilling CFG trajectories, and $\nu$-sampling for multi-step generation. While being the first large-scale full-band T2S distillation model in the sound community, SoundCTM-DiT-1B has demonstrated notable performance in both 1-step and multi-step generation.

## ETHICS STATEMENT

SoundCTM poses a risk for generating harmful or inappropriate content, offensive sound effects, or even sound content that infringes on copyrights. To mitigate these risks, it is crucial to carefully curate the training data as a first step. Furthermore, addressing these risks involves implementing robust content filtering, moderation mechanisms to prevent the creation of unethical, harmful sound contents.

## REPRODUCIBILITY STATEMENT

The source code is available at https://github.com/sony/soundctm. Moreover, we outline our training and sampling procedures in Algorithm 2 and Algorithm 1, and detailed implementation instructions for result reproducibility can be found in Appendix D.

## ACKNOWLEDGMENT

We sincerely acknowledge the support of everyone who made this research possible. Our heartfelt thanks go to Ganesh Nagaraja, Sachin M R, Srinidhi Srinivasa, and Giorgio Fabbro for their assistance. Computational resource of AI Bridging Cloud Infrastructure (ABCI) provided by National Institute of Advanced Industrial Science and Technology (AIST) was used.

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

# A    RELATED WORK

**Diffusion-based Models for T2S**    Recent work (Huang et al., 2023b;a; Liu et al., 2023; 2024b; Ghosal et al., 2023; Evans et al., 2024a;b;c) and competitions (Choi et al., 2023) report that LDM-based sound generation models outperform other approaches such as GANs. However, these LDM-based models suffer from slow inference speeds. To address this issue, two main approaches have been proposed for achieving faster generation in the sound domain. The first approach involves training distillation models, such as ConsistencyTTA (Bai et al., 2023) and AudioLCM (Liu et al., 2024c), using LDMs as teacher models. Both models are trained with the CD framework (Song et al., 2023). The second approach compresses long-form audio waveform using an encoder, as in the Stable Audio series (Evans et al., 2024a;b;c). For instance, the Stable Audio series compresses 95-second waveform signals into $z_0$, and its LDM is trained on this latent representation, unlike other LDM-based models that compress only 10-second audio signals with their encoder. SoundCTM is categorized as the distillation approach but based on CTMs in contrast to ConsistencyTTA and AudioLCM.

**Feature Extractor for Latent Distillation Models**    In the field of computer vision, several methods utilize pretrained feature extractors for LDM-based distillation (Sauer et al., 2024; Kang et al., 2024; Lin et al., 2024; Xu et al., 2024). Sauer et al. (2024); Lin et al. (2024); Xu et al. (2024) employ pretrained teacher models as a discriminator for adversarial diffusion distillation (Sauer et al., 2023). Kang et al. (2024) propose training LatentLPIPS, a latent counterpart of LPIPS, and using it to train GANs by distilling teacher LDMs.

In contrast to these approaches, we leverage pretrained teacher models for the CTM loss (not using GAN loss). Notably, none of these approaches have been explored in the sound domain.

**Training-free Controllable Generation**    Levy et al. (2023) demonstrate training-free controllable music generation using loss-based guidance (Yu et al., 2023). Their approach defines a task-specific loss, such as for music continuation and infilling, and incorporates the gradient of this loss into the inference sampling. Novack et al. (2024b;a) also present controllable generation by optimizing initial noisy latents $z_T$ based on loss computed in the target condition space. DITTO (Novack et al., 2024b) uses pretrained DMs and DITTO2 (Novack et al., 2024a) uses pretrained distillation models for $z_T$-optimization.

# B    DISCUSSION ON FEATURE EXTRACTOR CANDIDATES FOR CTM LOSS

When considering the use of external off-the-shelf pretrained networks as feature extractors, several potential candidates exist, such as VGGish (Hershey et al., 2017), PaSST (Koutini et al., 2022), LPAPS (Iashin and Rahtu, 2021), and CLAP (Wu* et al., 2023). However, these models are first trained with the data domain rather than the latent domain at not only different training data but also different sampling frequencies (16 kHz, 32kHz, and 48kHz, respectively), necessitating, for every training iteration, projecting $z_0$ to $x_0$ and resampling the audio from the training data's sampling frequency to that of the pretrained feature extractors. After resampling, the waveform must be transformed into another domain, such as the Mel-spectrogram domain, to match the input format of the extractors. Consequently, utilizing these models is not straightforward. This is why we propose to use the teacher's network as feature extractor for distillation loss.

# C    DISCUSSION ON INTEGRATING GAN LOSS FOR SOUNDCTM

Although we propose a new training framework for SoundCTM, the model cannot surpass the teacher model in 1-step generation, as achieved in the original CTM. To achieve such 1-step generation performance, integrating the GAN loss into Eq. (7) is required (Sauer et al., 2024; Xu et al., 2024; Lin et al., 2024).

However, obtaining performance improvements via the GAN loss requires careful selection of a discriminator. In fact, in our preliminary experiments with the 16kHz setting, even though we employed several off-the-shelf discriminators in the sound domain (gil Lee et al., 2023; Kumar et al., 2023; Iashin and Rahtu, 2021), they did not lead better performance than without using them. We

Table 6: Preliminary experiments with and without using GAN loss. Memory consumption is measured with a batch size of two per GPU.

| Method | # of steps | $FAD_{vgg} \downarrow$ | $KL_{passt} \downarrow$ | $IS_{passt} \uparrow$ | $CLAP \uparrow$ | Memory consumption for training |
|---|---|---|---|---|---|---|
| w/o. GAN | 2 | 2.43 | 1.29 | 7.50 | 0.42 | 46782 MiB |
| | 4 | 2.28 | 1.21 | 7.63 | 0.43 | |
| w/. GAN | 2 | 2.91 | 1.37 | 6.29 | 0.37 | 64492 MiB |
| | 4 | 2.37 | 1.35 | 7.53 | 0.41 | |

report one of the preliminary results of using GAN loss in Table 6. In this preliminary experiment, we use a multi-period discriminator (Kong et al., 2020) and a multi-band multi-scale complex STFT discriminator (Kumar et al., 2023) by following DAC (Kumar et al., 2023). Along with the lack of performance improvement from using the GAN loss, there are also significant increases in memory consumption. Therefore, in this work, we do not utilize the GAN loss. That said, developing new GAN setups including tailored for large-scale conditional sound generation might be worth exploring as future work.

# D    EXPERIMENTAL DETAILS

---

**Algorithm 2** SoundCTM's Training

**Require:** Probability of unconditional training $p_{uncond}$

1: **repeat**
2:     Sample $(\mathbf{x}_0, \mathbf{c}_{text})$ from $p_{data}$
3:     Calculate $\mathbf{z}_0$ through $\mathcal{E}(\mathbf{x}_0)$
4:     $\mathbf{c}_{text} \leftarrow \varnothing$ with $p_{uncond}$
5:     Sample $\boldsymbol{\epsilon} \sim \mathcal{N}(0, I)$
6:     Sample $t \in [0, T]$, $s \in [0, t]$, $u \in [s, t]$
7:     Sample $\omega \sim U[\omega_{min}, \omega_{max}]$
8:     Calculate $\mathbf{z}_t = \mathbf{z}_0 + t\boldsymbol{\epsilon}$
9:     Calculate $\texttt{Solver}(\mathbf{z}_t, \mathbf{c}_{text}, \omega, t, u; \boldsymbol{\phi})$
10:     Calculate $\mathbf{z}_{target}(\mathbf{z}_t, \mathbf{c}_{text}, \omega, t, u, s)$
11:     Calculate $\mathbf{z}_{est}(\mathbf{z}_t, \mathbf{c}_{text}, \omega, t, s)$
12:     Update $\boldsymbol{\theta} \leftarrow \boldsymbol{\theta} - \frac{\partial}{\partial \boldsymbol{\theta}} \mathcal{L}(\boldsymbol{\theta})$
13: **until** converged

---

**Algorithm 3** SoundCTM's Loss-based Guidance

**Require:** $\nu$, $\mathbf{c}_{text}$, $\mathbf{y}_{condition}$, $\omega$, $\gamma$, $\rho_{t_n}$

1: Start from $\mathbf{z}_{t_0}$
2: **for** $n = 0$ to $N - 1$ **do**
3:     $\tilde{t}_{n+1} \leftarrow \sqrt{1 - \gamma^2} t_{n+1}$
4:     Denoise $\mathbf{z}_{\tilde{t}_{n+1}} \leftarrow \nu G_{\boldsymbol{\theta}}(\mathbf{z}_{t_n}, \mathbf{c}_{text}, \omega, t_n, \tilde{t}_{n+1})$
5:         $+ (1 - \nu) G_{\boldsymbol{\theta}}(\mathbf{z}_{t_n}, \varnothing, \omega, t_n, \tilde{t}_{n+1})$
6:     $\mathbf{z}_{t_N | t_n} = G_{\boldsymbol{\theta}}(\mathbf{z}_{t_n}, \mathbf{c}_{text}, \omega, t_n, t_N)$
7:     $\hat{\mathbf{y}}_{condition} = f(\mathcal{D}(\mathbf{z}_{t_N | t_n}))$
8:     $\mathbf{z}_{\tilde{t}_{n+1}} = \mathbf{z}_{\tilde{t}_{n+1}} - \rho_{t_n} \nabla_{\mathbf{z}_{t_n}} \mathcal{L}(\hat{\mathbf{y}}_{condition}, \mathbf{y}_{condition})$
9:     Diffuse $\mathbf{z}_{t_{n+1}} \leftarrow \mathbf{z}_{\tilde{t}_{n+1}} + \gamma t_{n+1} \boldsymbol{\epsilon}$
10: **end for**
11: **Return** $\mathbf{z}_{t_N}$

---

## D.1    DETAILS OF T2S GENERATION ON 16KHZ

**Training Details**    We employ TANGO as the teacher diffusion model. For teacher's training, we use $\sigma_{data} = 0.25$ and time sampling $t \sim \mathcal{N}(-1.2, 1.2^2)$ by following EDM's training manner. We utilize the EDM's skip connection $c_{skip}(t) = \frac{\sigma_{data}^2}{t^2 + \sigma_{data}^2}$, output scale $c_{out}(t) = \frac{t\sigma_{data}}{\sqrt{t^2 + \sigma_{data}^2}}$, input scale $c_{in}(t) = \frac{1}{\sqrt{t^2 + \sigma_{data}^2}}$, and $c_{noise}(t) = \frac{1}{4} \ln t$ for $D_{\phi}$ modeling as

$$D_{\phi}(\mathbf{z}_t, t, \mathbf{c}_{text}) = c_{skip}(t)\mathbf{z}_t + c_{out}(t) F_{\phi}(c_{in}\mathbf{z}_t, c_{noise}(t), \mathbf{c}_{text}),$$

where $F_{\phi}$ refers to the actual neural network to be trained and the actual DSM loss is $\mathbb{E}_{\mathbf{z}_0, t, p_{0|t}(\mathbf{z}|\mathbf{z}_0)}[\lambda(t) \|\mathbf{z}_0 - D_{\phi}(\mathbf{z}, t, \mathbf{c}_{text})\|_2^2]$, where $\lambda(t) = \frac{t^2 + \sigma_{data}^2}{(t\sigma_{data})^2}$. Other than that, we follow the same network architecture, parameter size, and training setups as the original TANGO.

The network architecture of the teacher model and dataset for the teacher's training remained unchanged from the original ones. The network architecture of the teacher model is composed of the VAE-GAN (Liu et al., 2023), the HiFiGAN vocoder (Kong et al., 2020) as $\mathcal{D}$ and the Stable Diffusion UNet architecture (SD-1.5), which consists of 9 2D-convolutional ResNet (He et al., 2016) blocks as $D_{\phi}$. The UNet has a total of 866 M parameters, and the frozen FLAN-T5-Large text encoder (Chung et al., 2022) is used. The UNet employs 8 latent channels and a cross-attention dimension of 1024. During teacher model's training, we only train the UNet parts.

For student training, we mostly follow the original CTM's training setup. We utilize the EDM's skip connection $c_{\text{skip}}(t) = \frac{\sigma_{\text{data}}^2}{t^2 + \sigma_{\text{data}}^2}$ and output scale $c_{\text{out}}(t) = \frac{t\sigma_{\text{data}}}{\sqrt{t^2 + \sigma_{\text{data}}^2}}$ for $g_{\boldsymbol{\theta}}$ modeling as

$$g_{\boldsymbol{\theta}}(\mathbf{z}_t, \mathbf{c}_{\text{text}}, \omega, t, s) = c_{\text{skip}}(t)\mathbf{z}_t + c_{\text{out}}(t)\text{NN}_{\boldsymbol{\theta}}(\mathbf{z}_t, \mathbf{c}_{\text{text}}, \omega, t, s),$$

where $\text{NN}_{\boldsymbol{\theta}}$ refers to the actual neural network output. We initialize the student's $\text{NN}_{\boldsymbol{\theta}}$ with $\phi$ except for student model's $s$-embedding and $\omega$-embedding. The network architecture of the student model mostly follows that of the teacher model. However, since we inject time step $s$-embedding and $\omega$-embedding to the student network, we incorporate via auxiliary temporal embedding with positional embedding (Vaswani et al., 2017) add this embedding to the time step $t$-embedding. As for $\omega$-embedding, by following Meng et al. (2023), we apply Fourier embedding to $\omega$ and add this embedding to the time step $t$-embedding. In Eq. 9, we employ adaptive weighting with $\lambda_{\text{DSM}} = \frac{\|\nabla_{\boldsymbol{\theta}_L} \mathcal{L}_{\text{CTM}}^{\text{Sound}}(\boldsymbol{\theta};\phi)\|}{\|\nabla_{\boldsymbol{\theta}_L} \mathcal{L}_{\text{DSM}}^{\text{Sound}}(\boldsymbol{\theta})\|}$, where $\theta_L$ is the last layer of the student's network by following the original CTM.

We use $8\times$NVIDIA H100 (80G) GPUs and a global batch size of $64$ for the training. We choose $t$ and $s$ from the $N$-discretized timesteps to calculate $\mathcal{L}_{\text{CTM}}^{\text{Sound}}$. For $\mathcal{L}_{\text{DSM}}^{\text{Sound}}$ calculation, we opt to use $50\%$ of time sampling $t \sim \mathcal{N}(-1.2, 1.2^2)$. For the other half time, we first draw sample from $\xi \sim [0, 0.7]$ and transform it using $(\sigma_{\text{max}}^{1/\rho} + \xi(\sigma_{\text{min}}^{1/\rho} - \sigma_{\text{max}}^{1/\rho}))^\rho$. Throughout the experiments, for student training, we use $N = 40$, $\mu = 0.999$, $\sigma_{\text{min}} = 0.002$, $\sigma_{\text{max}} = 80$, $\rho = 7$, RAdam optimizer (Liu et al., 2020) with a learning rate of $8.0 \times 10^{-5}$, and $\sigma_{\text{data}} = 0.25$. We set the maximum number of ODE steps as 39 during training and we use Heun solver for $\texttt{Solver}(\mathbf{z}_t, \mathbf{c}_{\text{text}}, \omega, t, u; \phi)$. We also utilized TANGO's data augmentation (Ghosal et al., 2023, Sec. 2.3) during student training.

**Evaluation Details**    For large-NFE sampling, we follow the EDM's and the CTM's time discretization. Namely, if we draw $n$-NFE samples, we equi-divide $[0, 1]$ with $n$ points and transform it (say $\xi$) to the time scale by $(\sigma_{\text{max}}^{1/\rho} + (\sigma_{\text{min}}^{1/\rho} - \sigma_{\text{max}}^{1/\rho})\xi)^\rho$.

We use four objective metrics: the Frechet Audio Distance ($\text{FAD}_{\text{vgg}}$) (Kilgour et al., 2019) between the extracted embeddings by VGGish, the Kullback-Leibler divergence ($\text{KL}_{\text{passt}}$) between the outputs of PaSST (Koutini et al., 2022), a state-of-the-art audio classification model, the Inception Score ($\text{IS}_{\text{passt}}$) (Salimans et al., 2016) using the outputs of PaSST, and the CLAP score[7]. The lower FAD indicates better audio quality of the generated audio. The KL measures how semantically similar the generated audio is to the reference audio. The IS evaluates sample diversity. The CLAP score demonstrates how well the generated audio adheres to the given textual description.

### D.2    DETAILS OF FULL-BAND T2S GENERATION

**Network Architecture**    In this experiment, the teacher model is also based on LDM. A variational autoencoder (VAE) (Kingma and Welling, 2014) is used for the encoder and the decoder, compressing the $44.1$kHz monaural waveform to obtain the latent variable $\mathbf{z}_0$. We employ a fully-convolutional architecture, following the Descript Audio Codec (DAC) (Kumar et al., 2023), but without using the residual-vector quantizer. The encoder has $4$ layers, each of which downsamples the input audio waveform at rates $[4, 4, 8, 8]$. The decoder has $4$ corresponding layers, which upsample at rates $[8, 8, 4, 4]$. We set the encoder dimension to $64$ and the decoder dimension to $1536$. Thus, an overall data compression ratio of $16$ and the model has $63.5$M parameters.

For the VAE training, we mainly follow the DAC's training configuration. Since we remove the quantizer and trained a Gaussian VAE, we use a multi-scale Mel-loss, a feature matching loss, an adversarial loss, and a KL-regularization. To balance these four losses, the weighting factors are set to $15.0$, $2.0$, $1.0$, and $10^{-5}$, respectively. In terms of training dataset for the VAE part, we use AudioSet (Gemmeke et al., 2017) and MUSDB18-HQ (Rafii et al., 2019).

In contrast to TANGO-based SoundCTM, we employ DiT in this experiment. Specifically, our DiT architecture mostly follows Stable Audio-2.0 architecture (Evans et al., 2024b) but we remove the timing embedding since we do not aim at variable-length generation. For CLAP text embedding in the DiT, we use the "630k-audioset-best.pt" checkpoint from https://github.com/LAION-AI/CLAP. The model has $1.2$B parameters.

---

[7]We use the "630k-audioset-best.pt" checkpoint from https://github.com/LAION-AI/CLAP

Table 7: Semantic Preservation Evaluation of SoundCTM-DiT-1B on CLAP-LPAPS (Manor and Michaeli, 2024).

| Method | $\gamma$ | CLAP-LPAPS$\downarrow$ |
|---|---|---|
| VAE-reconstruction (Lower bound) | - | $2.55 \pm 0.39$ |
| Deterministic sampling | 0 | $\mathbf{3.34 \pm 0.74}$ |
| Stochastic sampling | 0.2 | $5.21 \pm 0.57$ |
| | 0.5 | $5.21 \pm 0.56$ |
| | 1.0 (CM-sampling) | $5.34 \pm 0.65$ |
| AC GT test samples | - | $6.07 \pm 0.41$ |

The network architecture of the student model mostly follows that of the teacher model. However, since we inject time step $s$-embedding and $\omega$-embedding to the student network, we incorporate $s$-information by using sinusoidal embeddings following the time step $t$-embedding of Stable Audio-2 and add this embedding to the time step $t$-embedding. As for $\omega$-embedding, by following Meng et al. (2023), we apply Fourier embedding to $\omega$ and add this embedding to the time step $t$-embedding. The model has $1.2$B parameters.

**Training Details**  For teacher training, we follow the EDM's variance exploding formulation. After obtaining $\mathbf{z}_0$ through the encoder, $\mathbf{z}_0$ is standardized globally to zero mean and standard deviation $\sigma_{\text{data}} = 0.5$ and we use $\sigma_{\text{data}} = 0.5$ and time sampling $t \sim \mathcal{N}(-1.2, 1.2^2)$ by following EDM2 (Karras et al., 2024). We utilize the EDM's skip connection $c_{\text{skip}}(t) = \frac{\sigma_{\text{data}}^2}{t^2 + \sigma_{\text{data}}^2}$, output scale $c_{\text{out}}(t) = \frac{t\sigma_{\text{data}}}{\sqrt{t^2 + \sigma_{\text{data}}^2}}$, input scale $c_{\text{in}}(t) = \frac{1}{\sqrt{t^2 + \sigma_{\text{data}}^2}}$, and $c_{\text{noise}}(t) = \frac{1}{4}\ln t$ for $D_\phi$ modeling as

$$D_\phi(\mathbf{z}_t, t, \mathbf{c}_{\text{text}}) = c_{\text{skip}}(t)\mathbf{z}_t + c_{\text{out}}(t)F_\phi(c_{\text{in}}\mathbf{z}_t, c_{\text{noise}}(t), \mathbf{c}_{\text{text}}),$$

where $F_\phi$ refers to the actual neural network to be trained and the actual DSM loss is $\mathbb{E}_{\mathbf{z}_0, t, p_{0|t}(\mathbf{z}|\mathbf{z}_0)}[\lambda(t)\|\mathbf{z}_0 - D_\phi(\mathbf{z}, t, \mathbf{c}_{\text{text}})\|_2^2]$, where $\lambda(t) = \frac{t^2 + \sigma_{\text{data}}^2}{(t\sigma_{\text{data}})^2}$.

For student training, we use $8\times$NVIDIA H100 (80G) GPUs and a global batch size of 192 for the training. We train 10K iterations. In Eq. 9, we employ adaptive weighting with $\lambda_{\text{DSM}} = \frac{\|\nabla_{\theta_L}\mathcal{L}_{\text{CTM}}^{\text{Sound}}(\theta;\phi)\|}{\|\nabla_{\theta_L}\mathcal{L}_{\text{DSM}}^{\text{Sound}}(\theta)\|}$, where $\theta_L$ is the last layer of the student's network by following the original CTM. We choose $t$ and $s$ from the $N$-discretized timesteps to calculate $\mathcal{L}_{\text{CTM}}^{\text{Sound}}$. For $\mathcal{L}_{\text{DSM}}^{\text{Sound}}$ calculation, we opt to use $50\%$ of time sampling $t \sim \mathcal{N}(-1.2, 1.2^2)$. For the other half time, we first draw sample from $\xi \sim [0, 0.7]$ and transform it using $(\sigma_{\max}^{1/\rho} + \xi(\sigma_{\min}^{1/\rho} - \sigma_{\max}^{1/\rho}))^\rho$. We use $N = 40$, $\mu = 0.96$, $\sigma_{\min} = 0.002$, $\sigma_{\max} = 80$, $\rho = 7$, RAdam optimizer (Liu et al., 2020) with a learning rate of $8.0 \times 10^{-5}$, $\omega \sim U[2.0, 5.0]$, and $\sigma_{\text{data}} = 0.5$. We set the maximum number of ODE steps as 39 during training and we use Heun solver for $\texttt{Solver}(\mathbf{z}_t, \mathbf{c}_{\text{text}}, \omega, t, u; \phi)$.

**Evaluation Details**  For large-NFE sampling, we follow the EDM's and the CTM's time discretization. Namely, if we draw $n$-NFE samples, we equi-divide $[0, 1]$ with $n$ points and transform it (say $\xi$) to the time scale by $(\sigma_{\max}^{1/\rho} + (\sigma_{\min}^{1/\rho} - \sigma_{\max}^{1/\rho})\xi)^\rho$.

### D.3 DETAILS OF SOUND INTENSITY CONTROL EXPERIMENT

We define $f(\mathbf{x}_0) := w * 20\log 10(\text{RMS}(\mathbf{x}_0))$, where $w$ represents the smoothing filter coefficients of a Savitzky-Golay filter (Savitzky and Golay, 1964) with a 1-second context window over the frame-wise value, and the RMS is the root mean squared energy of the generated sound. The target condition $\mathbf{y}_{\text{condition}}$ is a dB-scale target line or curve (See Figure 10 (a) and Figure 11 (a) for example).

We use 200 audio-text pairs from the AudioCaps testset for each of the 6 different types of $\mathbf{y}_{\text{condition}}$ as shown in Figure 10 (a) and Figure 11 (a).

For DITTO-2 framework on SoundCTM, we use Adam (Kingma and Ba, 2017) with a learning rate of 1.0. We also tested learning rates of $1.0 \times 10^{-1}$, $1.0 \times 10^{-2}$, and $5.0 \times 10^{-3}$ by following DITTO.

Table 8: Inference speeds measured on NVIDIA A100 with batch size of 1.

| Model | channels/sr | # of sampling | Inference speed [sec.] ↓ |
|---|---|---|---|
| **SoundCTM-DiT-1B (Ours)** | **1/44.1 kHz** | 1 | 0.141 |
| | | 2 | 0.193 |
| | | 4 | 0.298 |
| | | 8 | 0.501 |
| AudioLCM (Liu et al., 2024c) | 1/16 kHz | 1 | 0.169 |
| | | 2 | 0.181 |
| | | 4 | 0.220 |
| | | 8 | 0.224 |
| ConsistencyTTA (Bai et al., 2023) | 1/16 kHz | 1 | 0.216 |
| | | 2 | 0.253 |
| | | 4 | 0.306 |
| | | 8 | 0.472 |

Table 9: Subjective evaluation on AudioCaps testset on 16kHz setting. We report overall audio quality (OVAL) and text alignment (REL) with 95% Confidence Interval.

| Method | channels/sr | # of steps | $\omega$ | $\nu$ | OVAL ↑ | REL ↑ |
|---|---|---|---|---|---|---|
| Ground-truth | 1/16 kHz | - | - | - | $75.5 \pm 1.33$ | $74.9 \pm 1.50$ |
| AudioLCM (Liu et al., 2024c) | 1/16 kHz | 1 | 5.0 | - | $49.6 \pm 1.51$ | $56.9 \pm 1.67$ |
| | | 4 | 5.0 | - | $57.2 \pm 1.51$ | $64.1 \pm 1.59$ |
| **SoundCTM** | 1/16 kHz | 1 | 3.5 | 1.0 | $64.0 \pm 1.26$ | $69.4 \pm 1.49$ |
| | | 4 | 3.5 | 1.0 | $\mathbf{67.0 \pm 1.35}$ | $\mathbf{71.3 \pm 1.45}$ |

However, we cannot obtain better results than the case using $1.0$. The reason why we follow DITTO's setting instead of DITTO-2 is the learning rate setting is not provided in DITTO-2.

During $\mathbf{z}_T$-optimization, we use 1-step generation with $\omega = 3.5$ and $\nu = 1$. We perform 70 iterations for $\mathbf{z}_T$-optimization following DITTO's settings. After the optimization, we perform 16-step generation with $\omega = 3.5$ and $\nu = 2.0$. As suggested in DITTO-2 paper, we include results both using the deterministic sampling ($\gamma = 0$) and the stochastic sampling ($\gamma = 0.2$). For the time-dependent learning rate $\rho_t$ in SoundCTM's loss-based guidance framework, we use the overall gradient norm by following DITTO (Novack et al., 2024b, Section 5.4). For time discretization in multi-step generation, we use the same scheme as in T2S generation evaluation in Appendix 4.1.

# E  ADDITIONAL EXPERIMENTS

## E.1  NUMERICAL ANALYSIS FOR DETERMINISTIC SAMPLING

To objectively evaluate the semantic content preservation capability of SoundCTM in deterministic sampling, we calculate sample-wise reconstruction metrics, specifically LPAPS with the CLAP audio encoder (CLAP-LPAPS) (Manor and Michaeli, 2024) by using the toolkit[8]. The evaluation is conducted on 957 samples from the AudioCaps test set. We compute the CLAP-LPAPS between samples generated by SoundCTM-DiT-1B with 1-step and 16-step across several values of $\gamma$.

Furthermore, to benchmark these CLAP-LPAPS values, we also calculate scores for the following: 1). the ground truth audio samples from the AudioCaps test set and their corresponding reconstructed samples using the VAE in SoundCTM-DiT-1B (VAE-reconstruction), which reflects a signal-level reconstruction score; and 2). the ground truth audio samples from the AudioCaps test set and SoundCTM-DiT-1B's 1-step generated samples (AC GT test samples), which reflect a semantic-level random score within the same distribution.

We report the average and standard deviation of the CLAP-LPAPS scores in Table 7. Table 7 demonstrates that using deterministic sampling in SoundCTM preserves semantic meaning regardless of changes in the number of sampling steps.

---

[8] https://github.com/HilaManor/AudioEditingCode/tree/codeclean/evals.

Table 10: Performance comparisons between SoundCTM-UNet-1B and SoundCTM-DiT-1B on AudioCaps at full-band setting.

| Model | channels/sr | # of sampling steps | $\omega$ | $\nu$ | $FD_{openl3} \downarrow$ | $KL_{passt} \downarrow$ | $CLAP \uparrow$ |
|---|---|---|---|---|---|---|---|
| **Diffusion Models** | | | | | | | |
| UNet-Teacher w/. Heun Solver | 1/44.1kHz | 40 | 5.0 | - | 72.2 | 1.49 | 0.48 |
| DiT-Teacher w/. Heun Solver | 1/44.1kHz | 40 | 5.0 | - | 68.8 | 1.59 | 0.46 |
| **Distillation Models** | | | | | | | |
| SoundCTM-UNet-1B | 1/44.1kHz | 1 | 5.0 | 1.0 | 89.8 | 2.12 | 0.26 |
| | | 16 | 5.0 | 5.0 | 65.1 | 1.27 | 0.43 |
| **SoundCTM-DiT-1B** | 1/44.1kHz | 1 | 5.0 | 1.0 | 77.4 | 1.70 | 0.35 |
| | | 16 | 5.0 | 5.0 | 65.6 | 1.22 | 0.42 |

Table 11: Ablation study on loss function with SoundCTM-DiT-1B on AudioCaps testset at full-band setting.

| Model | # of sampling steps | $\omega$ | $\nu$ | $FD_{openl3} \downarrow$ | $KL_{passt} \downarrow$ | $CLAP \uparrow$ |
|---|---|---|---|---|---|---|
| SoundCTM-DiT-1B w/o. DSM loss | 1 | 5.0 | 1.0 | 77.9 | 1.74 | 0.35 |
| | 8 | 5.0 | 5.0 | 93.6 | 2.28 | 0.28 |
| | 16 | 5.0 | 5.0 | 108.7 | 2.60 | 0.23 |
| **SoundCTM-DiT-1B** | 1 | 5.0 | 1.0 | 77.4 | 1.70 | 0.35 |
| | 8 | 5.0 | 5.0 | 68.1 | 1.29 | 0.40 |
| | 16 | 5.0 | 5.0 | 65.6 | 1.22 | 0.42 |

### E.2 INFERENCE SPEED EVALUATION

We compare the inference speed of SoundCTM-DiT-1B with that of AudioLCM and ConsistencyTTA in Table 8. The inference speeds of all models are measured on an NVIDIA A100 with a batch size of 1. As shown in Table 8, we confirm that SoundCTM-DiT-1B can provide efficient trial-and-error processes not only in terms of the number of sampling steps but also with respect to inference time.

### E.3 SUBJECTIVE EVALUATION WITH SOUNDCTM-UNET ON 16KHZ SETTING

We conduct additional subjective evaluation for the 16 kHz setting. Especially, we focus on comparing 16 kHz models of AudioLCM and SoundCTM, given the numerical results in Table 2. We follow the same evaluation protocol that we use for the subjective evaluation in Section 4.2 (the full-band setting) and we ask 15 participants for the evaluation. As shown in Table 9, unlike the trends observed in $FAD_{vgg}$ in Table 2, SoundCTM outperforms AudioLCM, especially in terms of OVAL, even when comparing the results at 4 steps.

### E.4 ABLATION STUDIES ON SUB-MODULES

**Performance comparison between UNet and DiT on full-band settings**   We compare SoundCTM-UNet-1B and SoundCTM-DiT-1B on AudioCaps test set (957 samples) at full-band settings in Table 10. As shown in Table 10, SoundCTM-DiT-1B outperforms UNet-1B in 1-step generation. On the other hand, both models demonstrate comparable performance in 16-step generation.

**Loss function of DiT**   We provide an ablation study of SoundCTM-DiT-1B w/. and w/o. the DSM loss in Table 11. Note that the DSM loss serves to improve the accuracy of approximating the small jumps of conditional and unconditional trajectories during the training. As shown in Table 11, while the performance is comparable for 1-step sampling, incorporating the DSM loss leads to improved performance as the number of the sampling steps increases. This indicates the effectiveness of the DSM loss.

**Teacher's feature extractor of DiT**   We provide an ablation study of SoundCTM-DiT-1B w/. and w/o. the teacher's feature extractor $d_{teacher}$. As shown in Table 12, in contrast to UNet at 16 kHz case (see in Table 13), the performance for 1-step generation are comparable between w/. and w/o. $d_{teacher}$.

Table 12: Ablation study on teacher's feature extractor $d_{\text{teacher}}$ with SoundCTM-DiT-1B on AudioCaps testset at full-band setting.

| Model | # of sampling steps | $\omega$ | $\nu$ | $\text{FD}_{\text{openl3}} \downarrow$ | $\text{KL}_{\text{passt}} \downarrow$ | $\text{CLAP} \uparrow$ |
|---|---|---|---|---|---|---|
| SoundCTM-DiT-1B w/o. $d_{\text{teacher}}$ | 1 | 5.0 | 1.0 | 77.8 | 1.67 | 0.36 |
| | 8 | 5.0 | 5.0 | 79.6 | 1.58 | 0.35 |
| | 16 | 5.0 | 5.0 | 69.2 | 1.28 | 0.41 |
| **SoundCTM-DiT-1B** | 1 | 5.0 | 1.0 | 77.4 | 1.70 | 0.35 |
| | 8 | 5.0 | 5.0 | 68.1 | 1.29 | 0.40 |
| | 16 | 5.0 | 5.0 | 65.6 | 1.22 | 0.42 |

Table 13: Ablation study on teacher's feature extractor $d_{\text{teacher}}$ with SoundCTM on AudioCaps testset at 16kHz.

| Model | # of sampling steps | $\omega$ | $\nu$ | $\text{FAD}_{\text{vgg}} \downarrow$ | $\text{KL}_{\text{passt}} \downarrow$ | $\text{IS}_{\text{passt}} \uparrow$ | $\text{CLAP} \uparrow$ |
|---|---|---|---|---|---|---|---|
| SoundCTM w/o. $d_{\text{teacher}}$ | 1 | 3.5 | 1.0 | 2.45 | 1.28 | 6.83 | 0.42 |
| | 8 | 3.0 | 1.5 | 1.64 | 1.23 | 7.56 | 0.44 |
| | 16 | 3.0 | 2.0 | 1.62 | 1.23 | 7.16 | 0.44 |
| **SoundCTM** | 1 | 3.0 | 1.0 | 2.17 | 1.27 | 6.83 | 0.42 |
| | 8 | 3.0 | 1.5 | 1.51 | 1.21 | 8.08 | 0.46 |
| | 16 | 3.0 | 2.0 | 1.37 | 1.23 | 8.31 | 0.46 |

However, as the number of sampling steps increased, using feature extractor $d_{\text{teacher}}$ demonstrates better performance.

**Further ablations on $\nu$ and $\omega$ with SoundCTM-DiT-1B**  To further investigate the effects of $\nu$-sampling and $\omega$, we train student models by distilling with $\omega$ uniformly sampled from $[1.0, 7.0]$ during training. We evaluate AudioCaps test set $957$ samples in terms of $\text{FD}_{\text{openl3}}$ and CLAP scores as shown in Figure 12. From the 16-step results, we find that we can get better sample quality regardless of the $\omega$ value by $\nu$-sampling. On the other hand, examining the CLAP scores for 1-step generation reveals that the models might not be able to generate good samples for any value of $\nu$ when $\omega = 1$. This might highlight the important role of distilling the $\omega$-guided trajectory distillation, especially for 1-step generation.

### E.5 DOWNSTREAM APPLICATION: ZERO-SHOT BANDWIDTH EXTENSION

To further explore the zero-shot downstream application capabilities of SoundCTM, we apply it to a bandwidth extension task. Specifically, we used $500$ samples from the AudioCaps test set, applying a low-pass filter with a cut-off frequency of 3 kHz to generate observed low-passed signals by following the literature (Moliner et al., 2023). We perform zero-shot bandwidth extension on these observed signals with pretrained SoundCTM. We use the Log-spectral Distance (LSD) metric (Liu et al., 2024a; Wang and Wang, 2021) between the bandwidth extended signals and the ground-truth signals for the evaluation. For the zero-shot bandwidth extension algorithm, we apply the data consistency strategy (Moliner et al., 2023) to pretrained SoundCTM. As shown in Table 14 and Figure 13 demonstrate the potential zero-shot downstream application capabilities of SoundCTM for bandwidth extension task.

## F  LIMITATIONS

We propose using the teacher's network as the feature extractor for the distillation loss. However, performance improvements may be limited when the student is trained on different datasets than the teacher (e.g., training the student on a speech dataset using a teacher model trained on a music dataset).

As an initial step in exploring SoundCTM's capability for training-free controllable generation, we experimentally validate that our framework can sound intensity control using loss-based guidance.

Table 14: Zero-shot bandwidth extension experiments on pretrained SoundCTM

| Method | # of sampling steps | LSD $\downarrow$ |
|---|---|---|
| Low-passed (Observed signal) | - | 28.8 |
| Bandwidth extension with SoundCTM | 4 | 13.3 |
| | 8 | **12.9** |

However, it remains unclear whether this method is applicable to a wider range of downstream tasks, which we will address in future work.

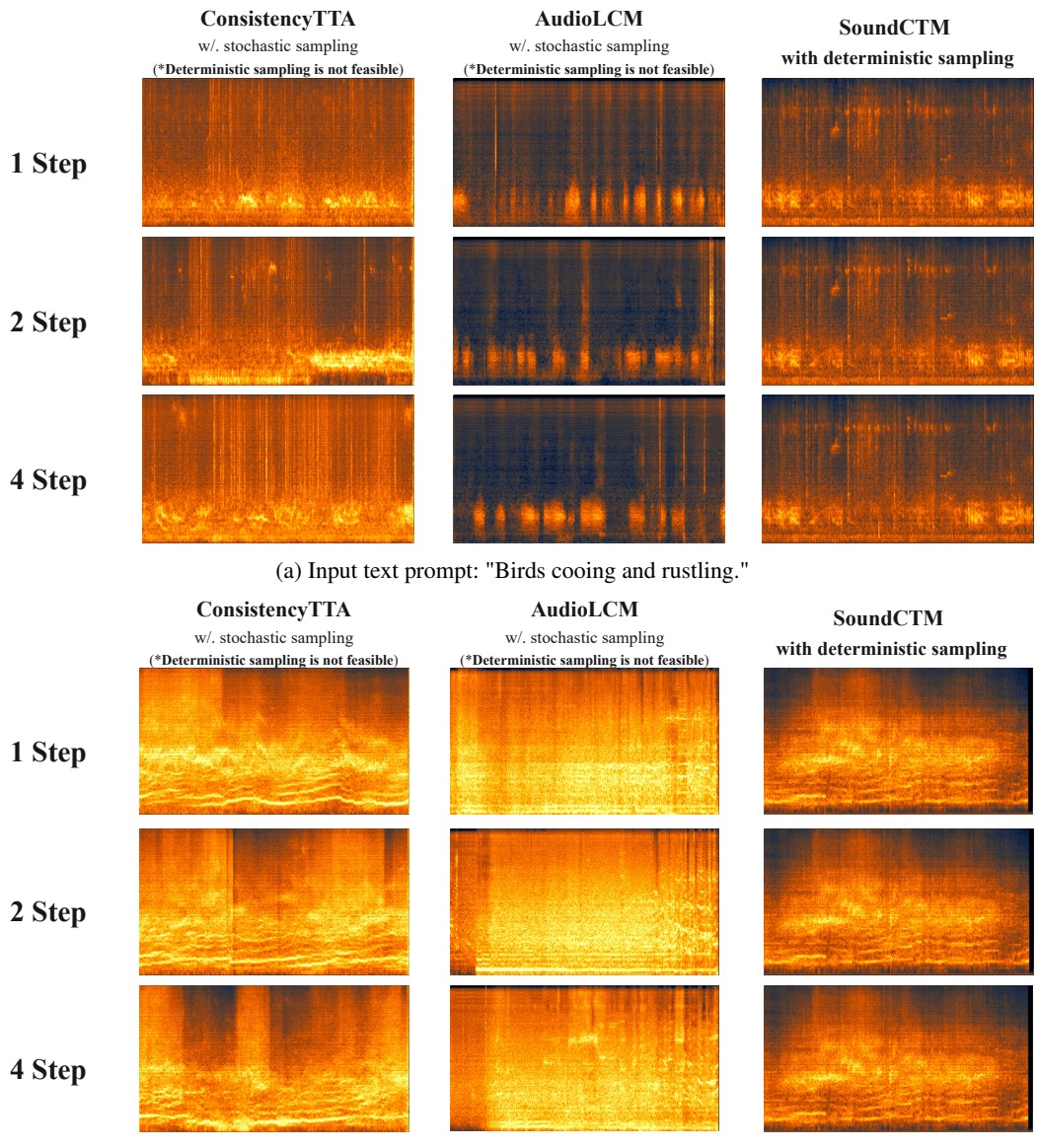

Figure 6: Visualization of spectrograms for generated samples using 1-step, 2-step, and 4-step generation with ConsistencyTTA (Bai et al., 2023), AudioLCM (Liu et al., 2024c), and SoundCTM. As ConsistencyTTA and AudioLCM, CD-based models (Song et al., 2023), inherently does not support deterministic sampling, the content of the generated samples cannot be preserved when varying the number of sampling steps, even when using the same initial noise and text prompts. This variability makes it challenging for users to control the output. In contrast, SoundCTM with deterministic sampling ($\gamma = 0$) is able to maintain consistent contents even when varying the number of sampling steps.

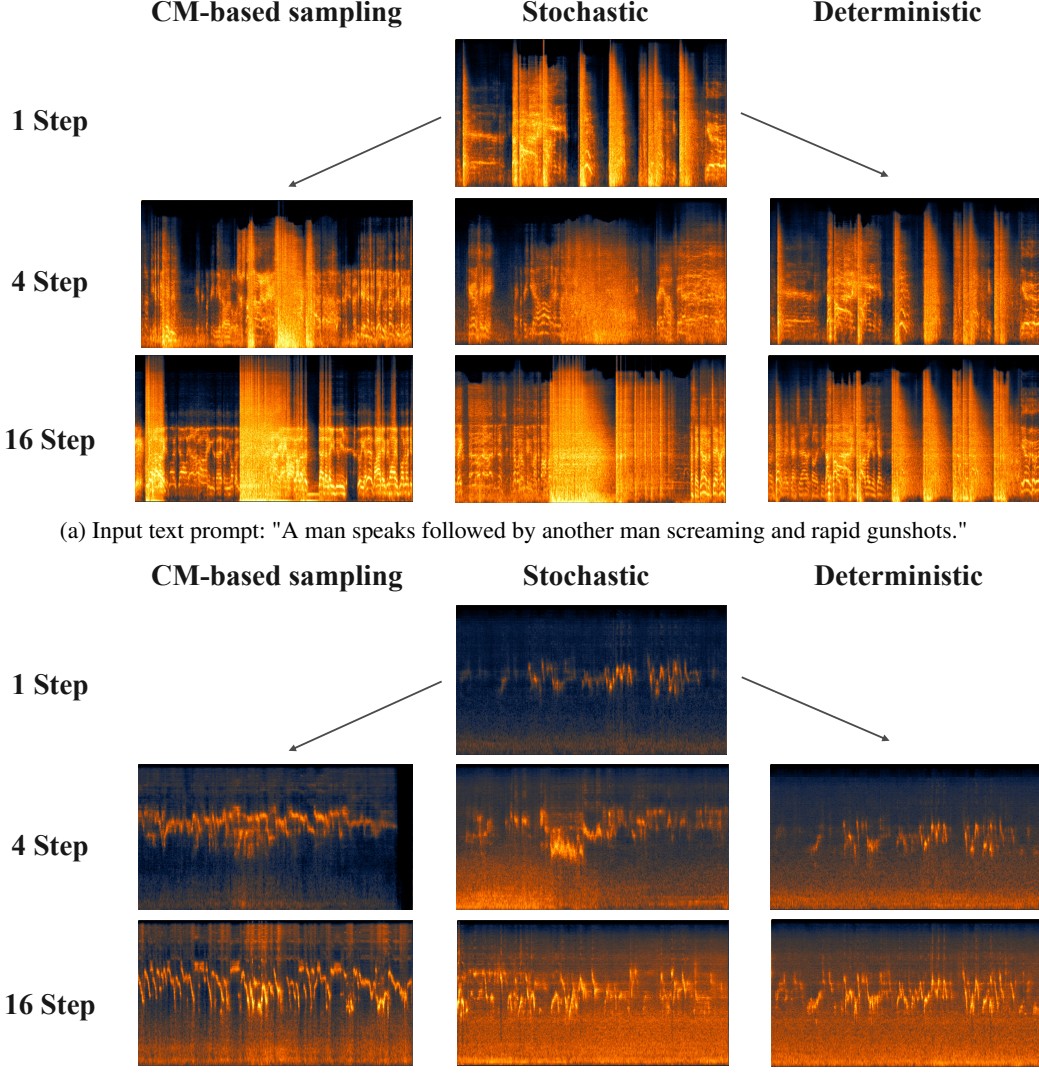

(a) Input text prompt: "A man speaks followed by another man screaming and rapid gunshots."

(b) Input text prompt: "White noise and then birds chirping."

Figure 7: Visualization of spectrograms for generated samples using 1-step, 4-step, and 16-step generation of SoundCTM-DiT-1B with CM-based sampling ($\gamma = 1$), stochastic sampling ($\gamma = 0.5$), and deterministic sampling ($\gamma = 0$).

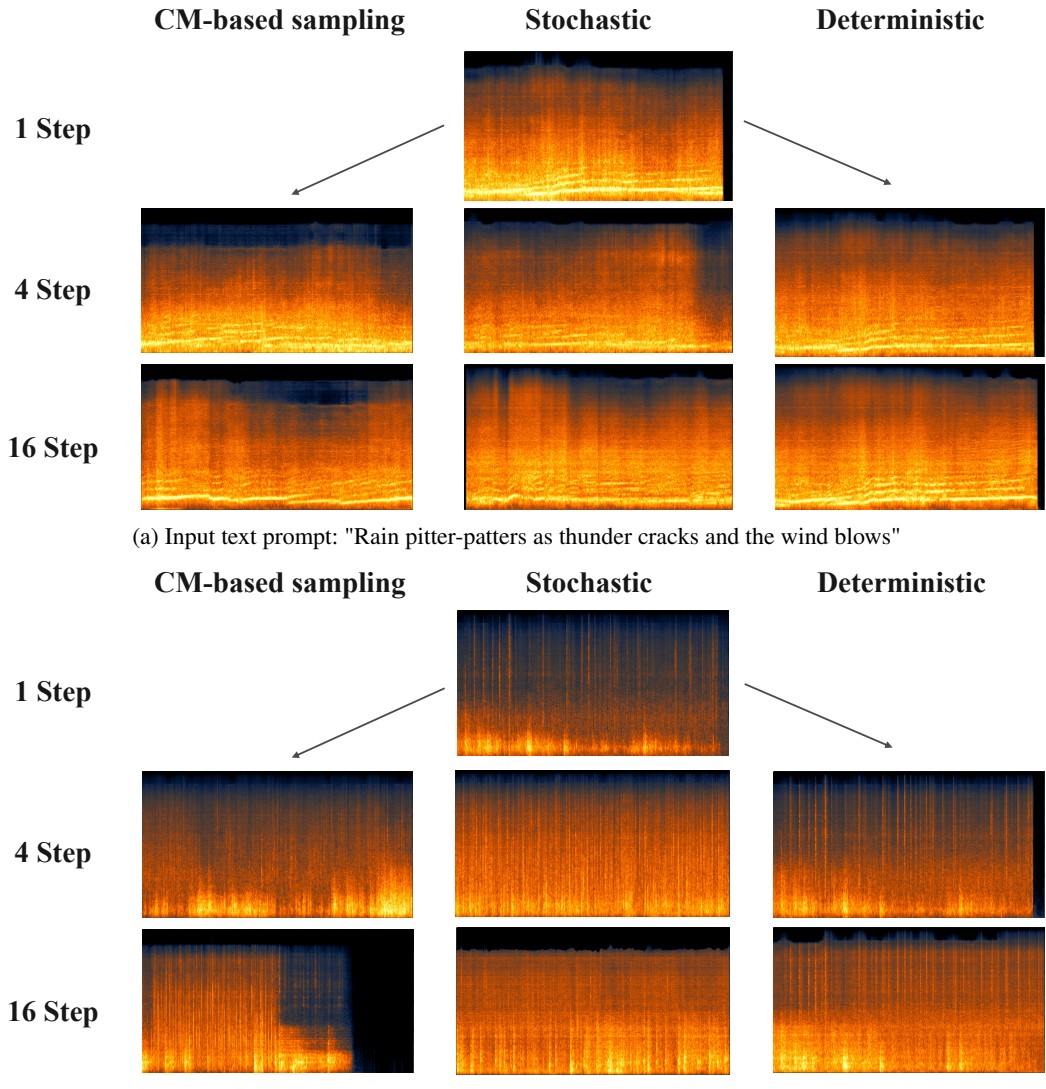

Figure 8: Visualization of spectrograms for generated samples using 1-step, 4-step, and 16-step generation of SoundCTM-DiT-1B with CM-based sampling ($\gamma = 1$), stochastic sampling ($\gamma = 0.5$), and deterministic sampling ($\gamma = 0$).

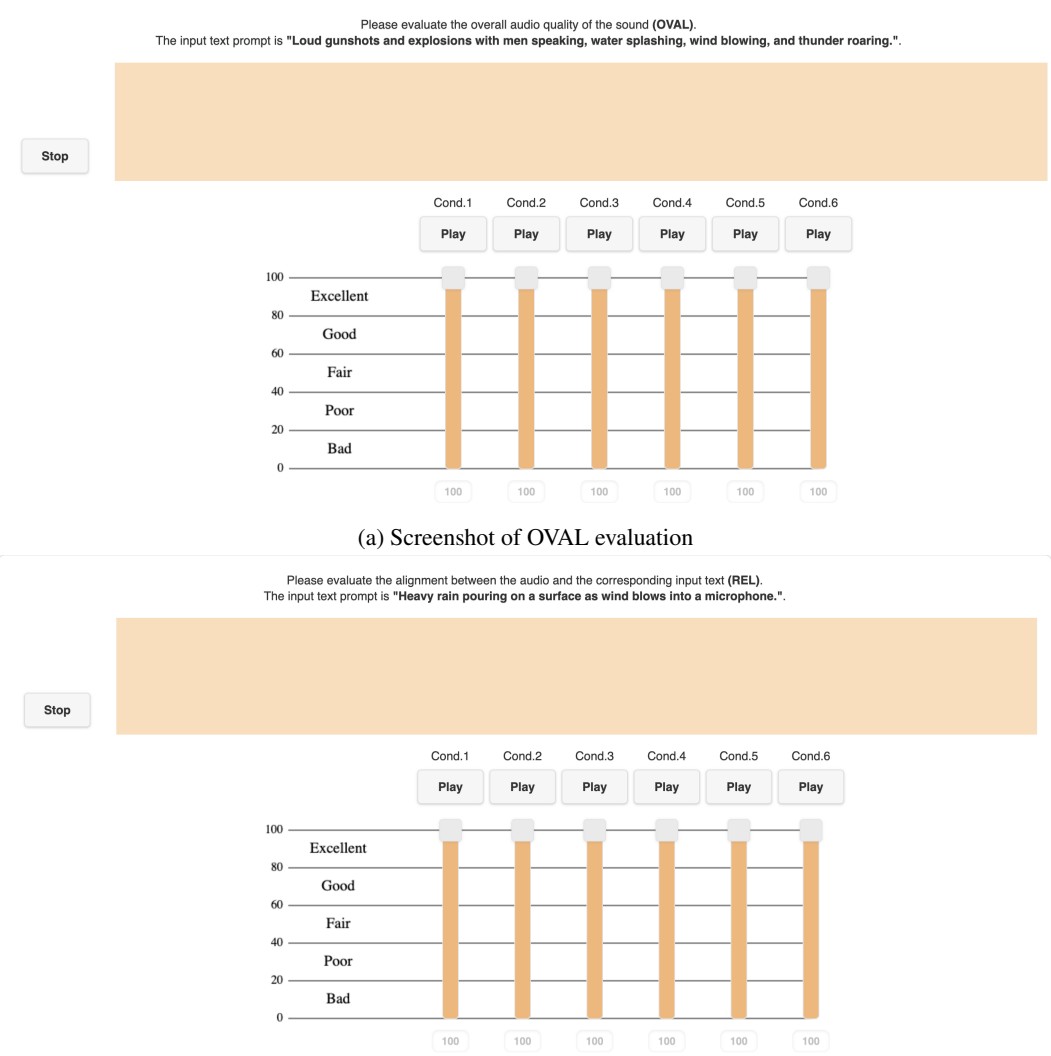

(a) Screenshot of OVAL evaluation

(b) Screenshot of REL evaluation

Figure 9: Screenshots of subjective evaluation.

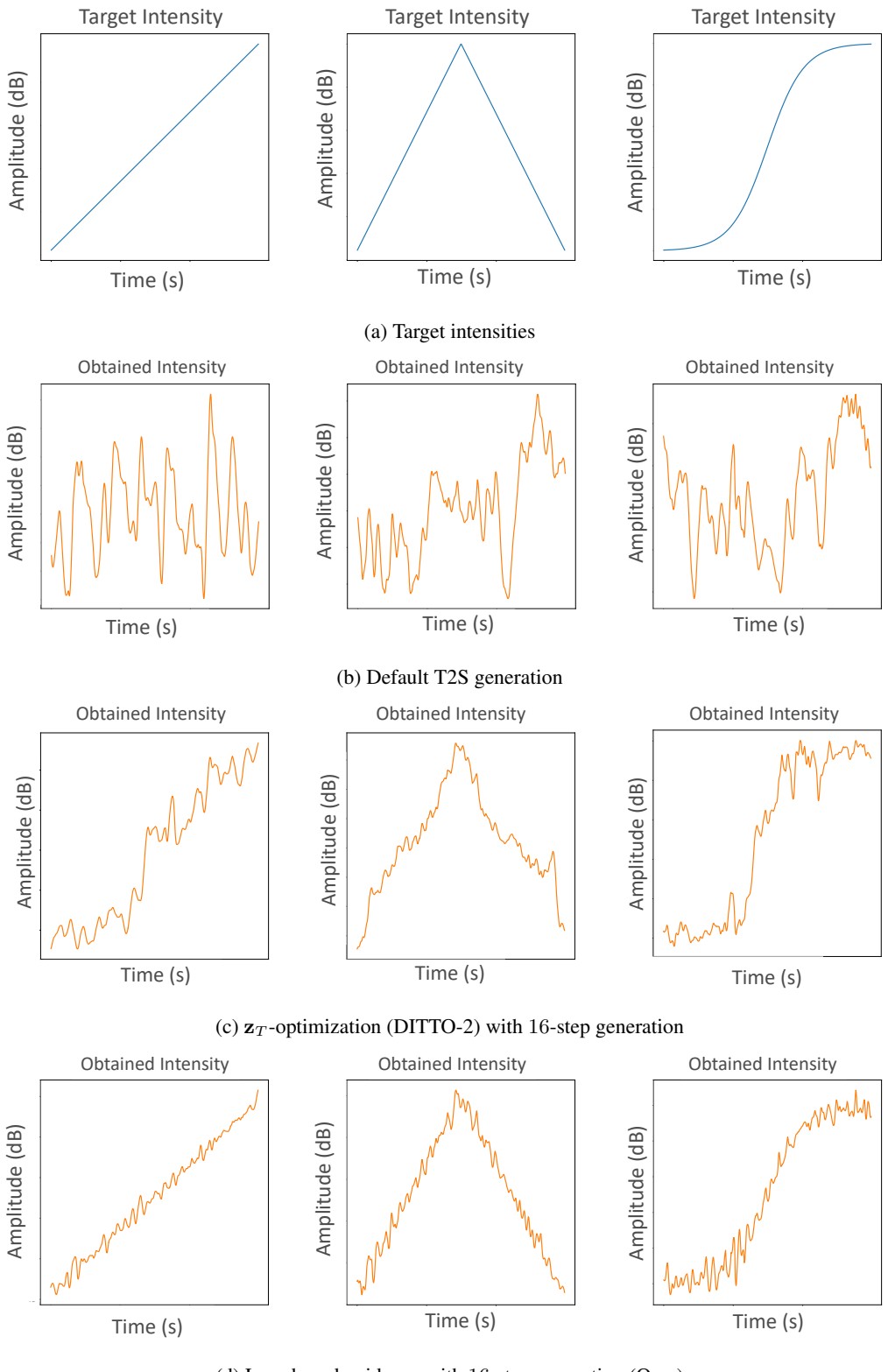

Figure 10: Target sound intensities and obtained intensities. We use the same text prompt within each column and different prompts across different columns. Note that we use 70 iterations for $\mathbf{z}_T$-optimization.

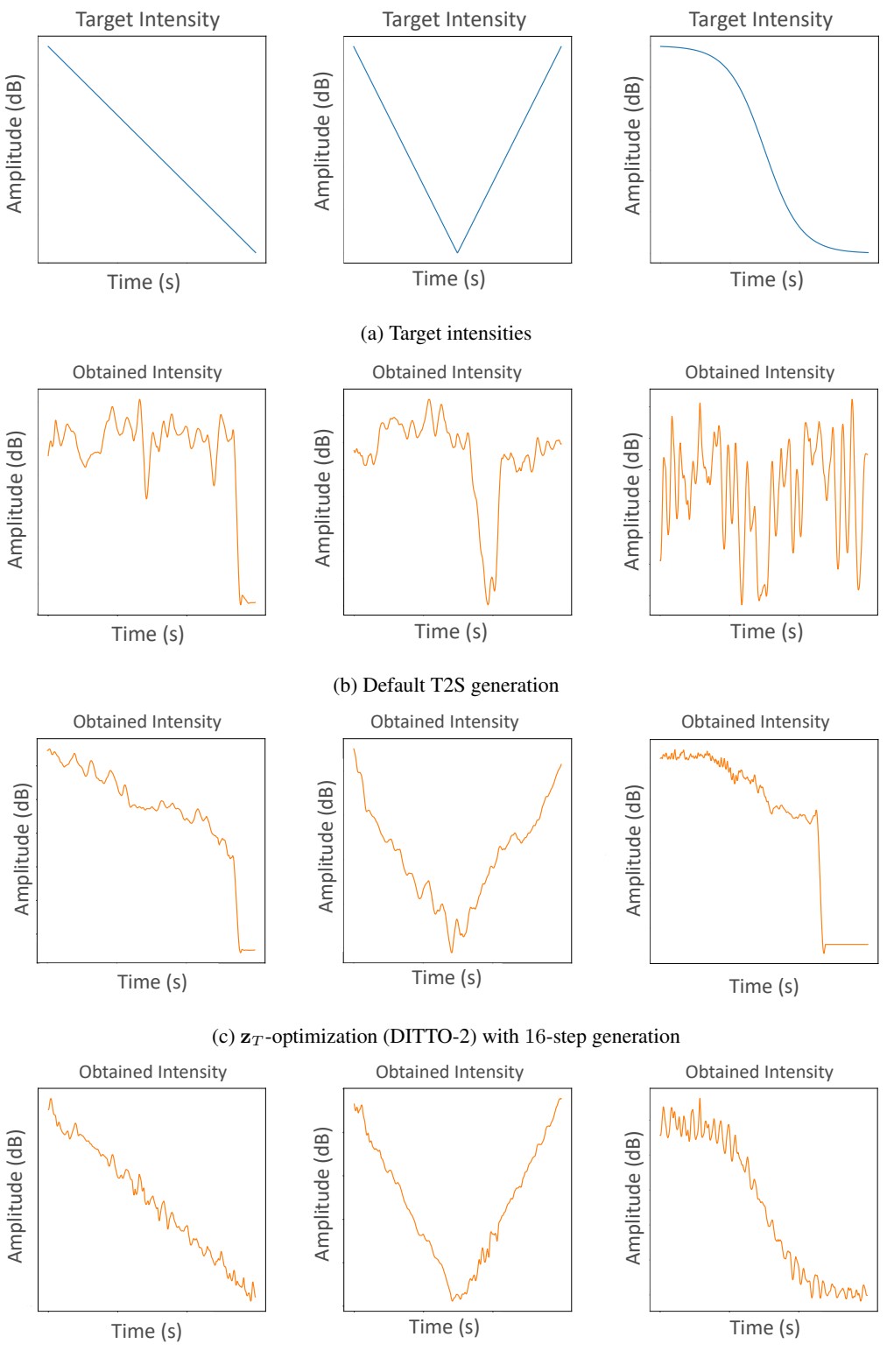

Figure 11: Target sound intensities and obtained intensities. We use the same text prompt within each column and different prompts across different columns. Note that we use 70 iterations for $\mathbf{z}_T$-optimization.

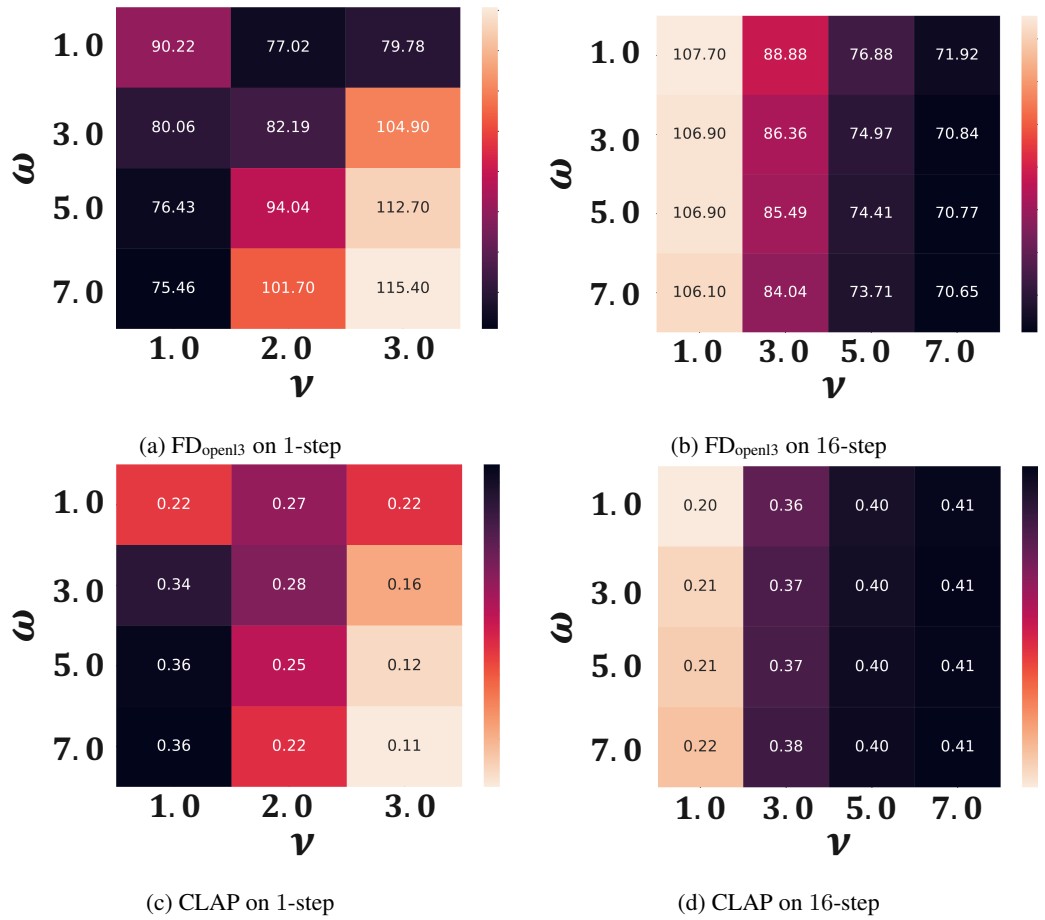

(a) FD$_{\text{openl3}}$ on 1-step

(b) FD$_{\text{openl3}}$ on 16-step

(c) CLAP on 1-step

(d) CLAP on 16-step

Figure 12: Influence of $\nu$ on SoundCTM-DiT-1B with 1- and 16-step sampling. Darker colors indicate better scores. In this case, during training, $\omega$ uniformly sampled from the range $[1.0, 7.0]$ in contrast to main paper.

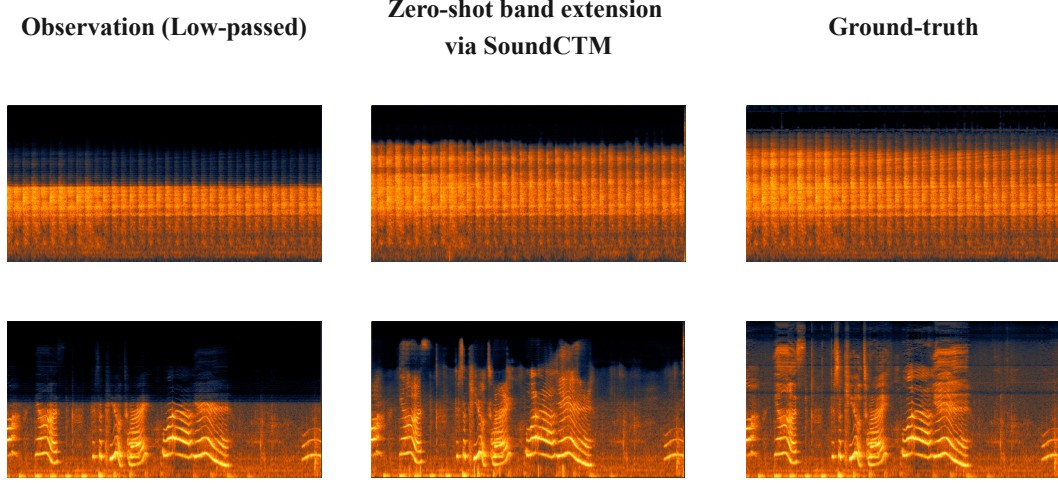

Figure 13: Visualization of spectrograms of low-passed signals, bandwidth-extended signals, and, ground-truth signals. Vertical axis is in log scale.

