# OpenReview forum: "SoundCTM: Unifying Score-based and Consistency Models for Full-band Text-to-Sound Generation"
_ICLR.cc/2025/Conference — ICLR 2025 Poster_

### Official Review · Reviewer_kUmE · 2024-11-02

**Soundness:** 3
**Presentation:** 3
**Contribution:** 3
**Rating:** 8
**Confidence:** 3

**Summary:**

This paper introduces SoundCTM, a novel approach to T2S generation that aims to enhance the efficiency and quality of sound creation in multimedia applications such as video games and films. By addressing the limitations of existing diffusion-based T2S models—which often suffer from slow inference speeds and subpar sample quality in 1-step generation—SoundCTM offers a flexible framework that allows for rapid trial-and-error through high-quality 1-step generation and subsequent refinement via multi-step deterministic sampling, all while preserving semantic content. The authors adapt the CTM training framework from computer vision, introducing a new feature distance metric that leverages the teacher network for distillation loss. They also propose a novel ν-sampling algorithm, enhancing the quality of multi-step generation. The model successfully scales up to one billion parameters, achieving promising results in both 1-step and multi-step full-band sound generation.

**Strengths:**

1. SoundCTM enables creators to quickly generate initial sound samples using high-quality 1-step generation and then refine these samples through multi-step deterministic sampling without altering the semantic content. This flexibility is highly valuable in practical sound design workflows.

2. By introducing a new feature distance metric that uses the teacher's network as a feature extractor for the distillation loss, the model enhances knowledge transfer from the teacher to the student. This approach leads to better performance compared to traditional L2 loss or external models, as it more effectively captures nuanced features relevant to sound generation.

3. The introduction of ν-sampling is a significant contribution. By combining conditional and unconditional student models, ν-sampling improves the quality of multi-step generation and provides another avenue for quality enhancement.

4. The ability to maintain semantic consistency across different sampling steps is crucial for practical applications. SoundCTM effectively addresses this by enabling deterministic sampling, which is essential for sound designers who require consistent outputs during refinement.

**Weaknesses:**

1. While the authors mention that preliminary experiments with GAN loss did not yield improvements, they do not deeply explore alternative strategies or provide a thorough analysis of why GAN loss integration was unsuccessful. A more detailed investigation could potentially uncover methods to enhance the model's performance, especially in 1-step generation.

2. Relying on the teacher's network as a feature extractor may limit the model's flexibility, particularly if the teacher and student models are trained on different datasets or domains. This dependency could pose challenges when attempting to generalize the model to new or diverse datasets.

3. The paper briefly demonstrates the application of SoundCTM to downstream tasks like controllable generation but does not explore this in depth. A more comprehensive study on various applications could showcase the versatility of the model.

4. The paper does not provide a comparison between SoundCTM and flow matching approaches. Such a comparison could offer valuable insights into the advantages or limitations of the proposed method relative to other leading techniques in the field.

**Questions:**

1. Do the authors plan to explore alternative GAN architectures or training strategies to effectively integrate GAN loss and potentially enhance the quality of 1-step generation? Understanding the obstacles faced and possible solutions could be beneficial for future work.

2. How does SoundCTM perform when the teacher and student models are trained on different datasets? Additionally, how well does the model adapt to other audio domains such as music generation or speech synthesis? Insights into its generalizability would be valuable.

3. Can the authors provide more detailed insights into the ν-sampling algorithm? Specifically, how does ν-sampling impact the model's performance compared to standard sampling methods, and what are the theoretical underpinnings that justify its effectiveness?

4. Have the authors considered using external pretrained networks as feature extractors for the distillation loss? If so, how might this affect the model's performance and flexibility, especially in scenarios where the teacher network is not available or differs significantly from the student?

5. How does SoundCTM compare with flow matching methods in terms of performance, computational efficiency, and ease of implementation? Are there specific advantages in choosing SoundCTM over flow matching approaches for text-to-sound generation tasks?

6. Beyond text-to-sound generation, can SoundCTM be applied to other audio-related tasks such as speech synthesis, audio style transfer, or sound enhancement? What adaptations would be necessary, and how might it perform compared to specialized models designed for those tasks?

---

> ### Author Response · Authors · 2024-11-22
>
> We appreciate the reviewer's dedication and the valuable feedback they have provided. In this response, we have taken care to address each of their questions and concerns to polish up our work.
>
> #### **Response to W1 and Q1**
> >**"W1. While the authors mention that preliminary experiments with GAN loss did not yield improvements, they do not deeply explore alternative strategies or provide a thorough analysis of why GAN loss integration was unsuccessful. A more detailed investigation could potentially uncover methods to enhance the model's performance, especially in 1-step generation."**
>
> >**"Q1. Do the authors plan to explore alternative GAN architectures or training strategies to effectively integrate GAN loss and potentially enhance the quality of 1-step generation? Understanding the obstacles faced and possible solutions could be beneficial for future work."**
>
> We appreciate your suggestion regarding the integration of GAN loss into the current version of SoundCTM. We recognize this as one of the most significant items in our future work. In this response, we will first explain why we deprioritized GAN integration in this paper and then outline our future plans for its implementation.
>
>
>  **[Prioritized Efforts on Generator]**
> To successfully integrate GAN loss, it is essential to design an appropriate discriminator architecture and find hyperparameters tailored to combinations of the generator and discriminator. In preliminary experiments, as we mentioned in the paper, we attempted to incorporate GAN loss but encountered failures. Based on this, we decided to **focus on the core component of SoundCTM—the generator—rather than further exploring discriminator designs at this stage**. Notably, the proposed methods in this paper will be open-sourced upon acceptance, including the training/inference codebase and checkpoints. We believe this will be also a contribution to the audio/sound research community.
>
> **[Our potential future plan for GAN integration]**
> Regarding our future work on discriminator design, we plan to compute GAN loss in the latent domain rather than the waveform domain, as was done in our preliminary experiments, inspired by succeeded approaches such as LADD [1]. One example of a possible discriminator architecture draws from our training framework and successful recipes in the computer vision domain [1,2]. Specifically, we are considering leveraging a pretrained frozen teacher network (e.g., DiT) combined with small-scale trainable discriminator heads.
> A suitable design for these heads might involve architectures capable of handling 1D sequences, similar to those introduced in StyleGAN-T [3].
>
> It is important to note that merely applying this idea does not guarantee that the GAN loss will work. Several challenges remain, including hyperparameter tuning (e.g., balancing GAN loss with other losses, a warm-up strategy for GAN loss, and configuring the optimizer) and potential further modifications to the discriminator architecture.
>
> [1] Sauer, A., Boesel, F., Dockhorn, T., Blattmann, A., Esser, P. and Rombach, R.. Fast high-resolution image synthesis with latent adversarial diffusion distillation. SIGGRAPH Asia, 2024
>
> [2] Kim, D., Lai, C.H., Liao, W.H., Murata, N., Takida, Y., Uesaka, T., He, Y., Mitsufuji, Y. and Ermon, S.. Consistency trajectory models: Learning probability flow ode trajectory of diffusion. ICLR 2024
>
> [3] Sauer, A., Karras, T., Laine, S., Geiger, A. and Aila, T.. Stylegan-t: Unlocking the power of gans for fast large-scale text-to-image synthesis. ICML 2023

---

> ### Author Response · Authors · 2024-11-22
>
> #### **Response to W2 and Q2**
> >**"W2. Relying on the teacher's network as a feature extractor may limit the model's flexibility, particularly if the teacher and student models are trained on different datasets or domains. This dependency could pose challenges when attempting to generalize the model to new or diverse datasets."**
>
> >**"Q2. How does SoundCTM perform when the teacher and student models are trained on different datasets? Additionally, how well does the model adapt to other audio domains such as music generation or speech synthesis? Insights into its generalizability would be valuable."**
>
> The limitations related to dataset generalizability in teacher and student models represent a common challenge for distillation-based generative models [4,5,6], rather than an issue specific to our feature extractors. To ensure clarity, we first discuss this topic from the perspective of distillation models and then from the perspective of feature extractors.
>
> **[Clarification: Distillation model viewpoint]**
>
> Distillation-based generative models typically rely on the assumption that student models are distilled based on the data distribution learned by teacher models. Consequently, the training dataset for student models should be a subset of the training dataset for teacher models.
>
> For example, it is within the scope of current frameworks to distill a student model on a new dataset of general sound using a teacher model trained with music, speech, and general sound data. However, it falls outside the scope to train a student model on speech data with a teacher model trained exclusively on instrumental music data.
>
> Consequently, in SoundCTM and other distillation-based models, applying a student model to tasks like music generation or speech synthesis requires retraining or finetuning the teacher model on the target dataset if the teacher model was trained on a dataset that does not include such data. Addressing this limitation represents a meaningful direction for future research.
>
> **[Feature extractor viewpoint]**
>
> Here, we discuss the flexibility of teacher feature extractors in cases where the student model's dataset is a subset of the teacher model's data distribution.
>
> For instance, consider a scenario where the teacher model is trained on music, speech, and general sound data, while the student model is trained on a new dataset, such as game sound effects.
> In such scenarios, it is expected that effective distillation can be achieved using our proposed feature extractor, as described in Eq. (5). However, for certain samples in the new dataset, L2 loss might fail to provide effective gradients. In such cases, employing robust losses like L1 loss or Pseudo-Huber loss [7] could potentially enable effective distillation with our feature extractor.
>
> This line of study should be a comprehensive investigation across different combinations of teacher models, student models, and datasets. While this is beyond the scope of our single paper, it is an interesting direction for future research.
>
> [4] Song, Y., Dhariwal, P., Chen, M. and Sutskever, I.. Consistency models. ICML 2023.
>
> [5] Liu, X., Gong, C. and Liu, Q., 2022. Flow straight and fast: Learning to generate and transfer data with rectified flow. ICLR 2023.
>
> [6] Sauer, A., Lorenz, D., Blattmann, A. and Rombach, R.. Adversarial diffusion distillation. ECCV 2024.
>
> [7] Song, Y., Dhariwal, P.. Improved techniques for training consistency models. ICLR 2024.

---

> ### Author Response · Authors · 2024-11-22
>
> #### **Response to W3**
>
> >**"W3. The paper briefly demonstrates the application of SoundCTM to downstream tasks like controllable generation but does not explore this in depth. A more comprehensive study on various applications could showcase the versatility of the model."**
>
> **[Another example of downstream applications]**
> As you noted, our exploration of downstream task applications was not conducted in depth. However, as elaborated in a later reply (see "Response to Q6"), while there are opportunities to apply SoundCTM to various downstream tasks, these applications are not trivial to be solved by simple plug-and-play or fully covered within a single paper. We anticipate that future research will delve further into the applicability of SoundCTM to diverse downstream tasks.
>
> To foster follow-up studies, we provide an additional example of a zero-shot application to a downstream task done in this rebuttal period. We present the results of applying SoundCTM to the bandwidth extension task in Appendix D.5.
>
> #### **[Experimental setup for bandwidth extension]**
>
> - Task: Bandwidth extension of low-pass filtered signals using pretrained SoundCTM in a zero-shot setting. Cut-off frequency is 3 kHz.
> - Test Data: 500 samples from the AudioCaps test set.
> - Evaluation Metric: Log-spectral Distance (LSD) [8,9] calculated against ground-truth data.
>
> |Method|steps|LSD $\downarrow$|
> |----|----|----|
> |Low-passed (observation)|-|28.8|
> |Band extenson with SoundCTM|4|13.3|
> ||8|**12.9**|
>
> This result briefly demonstrates the applicability of SoundCTM in bandwidth extension task. For further details and spectrogram visualization examples, please refer to Figure 12 and Appendix D.5. Additionally, for the public good, we will make the code for sound intensity control and band extension open-sourced upon acceptance.
>
> [8] H. Liu, K. Chen, Q. Tian, W. Wang and M. D. Plumbley. AudioSR: Versatile Audio Super-Resolution at Scale. ICASSP 2024.
>
> [9] H. Wang and D. Wang, 2021. Towards Robust Speech Super-Resolution. in IEEE/ACM Trans. on ASLP, vol. 29, pp. 2058-2066, 2021.

---

> ### Author Response · Authors · 2024-11-22
>
> #### **Response to W4 and Q5**
>
> >**"W4. The paper does not provide a comparison between SoundCTM and flow matching approaches. Such a comparison could offer valuable insights into the advantages or limitations of the proposed method relative to other leading techniques in the field."**
>
> >**"Q5. How does SoundCTM compare with flow matching methods in terms of performance, computational efficiency, and ease of implementation? Are there specific advantages in choosing SoundCTM over flow matching approaches for text-to-sound generation tasks?"**
>
> Flow-matching methods generally include 'vanilla' flow-matching [10] and rectified flow [5], which follows the Reflow approach. Below, we describe how these two methods differ from SoundCTM, followed by a discussion of their performance, computational efficiency, and ease of implementation.
>
> **[Vanilla flow-matching is not distillation model]**
> Vanilla flow-matching is not a distillation model and should therefore be compared at the level of the teacher diffusion model. Experimentally, in audio field, it has been shown that vanilla flow-matching does not yield reasonable samples in a 1-step generation even similar experimental setup with ours, as demonstrated in LAFMA [11]. Additionally, since vanilla flow-matching can be interpreted as an ODE-based modeling, it could be possible to extend SoundCTM's idea to distill such models [12].
>
> **[Reflow can be compared with distillation model]**
> Reflow shares the same distillation model framework as SoundCTM.
> The primary distinction is in how they handle the teacher model's ODE trajectory. SoundCTM explicitly learns jumps between points along the PF-ODE trajectory, allowing for anytime-to-anytime jumps. On the other hand, Reflow straightens the trajectory via distillation, obtaining a new trajectory (velocity) that forms a straight line between $t=T$ and $t=0$.
>
> **[Practical differences between SoundCTM and Reflow-based models from performance, computational efficiency, and ease of implementation]**
>
> Clearly concluding the upside and downside of SoundCTM and Reflow-based models in terms of performance, computational efficiency, and ease of implementation is far beyond the scope of a single paper.
>
> Below, we outline some of the key factors that require further exploration:
> - Performance: Evaluating performance necessitates considering scalability from both the perspectives of model size and training data.
> - Computational efficiency: For SoundCTM, computational efficiency depends on factors such as the choice of $\text{Solver}$ and the number of $\text{Solver}$ steps during training. In Reflow-based models, efficiency is influenced by the number of Reflow processes required, the size of the training data and model, and the total number of function evaluations (NFE) needed to generate offline noise and sample coupling data. For example, FlashAudio [13], a Reflow-based model in concurrent work, involves preparing a pretrained vanilla flow-matching model, conducting two rounds of Reflow training, and performing a one-step distillation. This amounts to three training processes (excluding pretrained model preparation) and two rounds of coupling data generation.
> - Ease of implementation: Implementation complexity is influenced by additional techniques aimed at improving performance and computational efficiency.
>
> Taking these considerations into account, we recognize the need for further exploration to assess the upsides and downsides of these methods. We believe that such further exploration would provide meaningful insights for the sound community.
>
> [10] Lipman, Y., Chen, R.T., Ben-Hamu, H., Nickel, M. and Le, M.. Flow matching for generative modeling. ICLR 2023.
>
> [11] Guan, W., Wang, K., Zhou, W., Wang, Y., Deng, F., Wang, H., Li, L., Hong, Q. and Qin, Y.. LAFMA: A latent flow matching model for text-to-audio generation. Interspeech 2024.
>
> [12] Yang, L., Zhang, Z., Zhang, Z., Liu, X., Xu, M., Zhang, W., Meng, C., Ermon, S. and Cui, B., 2024. Consistency flow matching: Defining straight flows with velocity consistency. arXiv preprint arXiv:2407.02398.
>
> [13] Liu, H., Wang, J., Huang, R., Liu, Y., Lu, H., Xue, W. and Zhao, Z., 2024. FlashAudio: Rectified flows for fast and high-fidelity text-to-audio generation. arXiv preprint arXiv:2410.12266.

---

> ### Author Response · Authors · 2024-11-22
>
> #### **Response to Q3**
>
> >**"Q3. Can the authors provide more detailed insights into the ν-sampling algorithm? Specifically, how does ν-sampling impact the model's performance compared to standard sampling methods, and what are the theoretical underpinnings that justify its effectiveness?"**
>
> $\nu$-sampling particularly lead better performance when the number of sampling steps increases. Please refer to Figure 12 and Appendix D.4.. Compared to standard diffusion-based sampling methods, $\nu$-sampling plays a role in consistency-trajectory "distillation" models akin to that of CFG in "diffusion" models. (Thus, intuitively, as illustrated in Figure 13, it helps alleviate ‘blurriness’ in the generated outputs.)
>
> The theoretical underpinnings of $\nu$-sampling’s effectiveness with a larger number of sampling steps can be intuitively explained as follows:
> - When conducting small jump on the "CFG-guided" trajectory (multi-step generation), the DSM loss (Eq. (8)) introduces estimated errors in small jumps on the "CFG-guided" trajectory. This is because the DSM loss in Eq. (8) does not learn the "CFG-guided" score. (Whether replacing Eq. (8) with a "CFG-guided" score would be optimal remains open for further discussion.)
> - The unconditional student model used in $\nu$-sampling compensates for these errors, functioning similarly to CFG-sampling in diffusion models.
>
> Below, we provide additional explanations for potential concerns:
> - Do we still need DSM loss? => Yes, without DSM loss, the small jumps of the conditional/unconditional student model become inaccurate, and one downside of this is that the benefits of $\nu$-sampling cannot be realized. (See Table 11.)
> - Do we still need to distill CFG-trajectory? => Yes, without this, even using $\nu$-sampling for 1-step generation, we cannot get good samples (see $\omega=1$ in Figure 12 (c).).
>
>
> #### **Response to Q4**
> >**"Q4. Have the authors considered using external pretrained networks as feature extractors for the distillation loss? If so, how might this affect the model's performance and flexibility, especially in scenarios where the teacher network is not available or differs significantly from the student?"**
>
> To the best of our knowledge, no pretrained external network exists for the latent space $z$ that we aim to distill. We have discussed the possibility of using pretrained external networks in the data space in Appendix B, and we would appreciate it if you could refer to it for more information.

---

> ### Author Response · Authors · 2024-11-22
>
> #### **Response to Q6**
> >**"Q6. Beyond text-to-sound generation, can SoundCTM be applied to other audio-related tasks such as speech synthesis, audio style transfer, or sound enhancement? What adaptations would be necessary, and how might it perform compared to specialized models designed for those tasks?"**
>
> It is important to note that SoundCTM's architecture is not originally designed for other audio-related tasks such as speech synthesis, audio style transfer, or sound enhancement.
> Based on this premise, we discuss the potential for applying SoundCTM to such tasks in two contexts:
> - Distillation of specialized diffusion models
> - Extension of SoundCTM’s zero-shot capabilities.
>
> **[We can distill specialized diffusion models via SoundCTM's training framework]**
> One possible application is the acceleration of task-specific conditional diffusion models by distilling specialized diffusion-based models designed for individual tasks. This involves utilizing SoundCTM's core idea of learning anytime-to-anytime jumps. For instance, applying SoundCTM's training approach to distill SGMSE [14] or UNIVERSE [15], a conditional diffusion model for speech enhancement, could serve as an example of this strategy.
>
> **[Zero-shot capability for other tasks]**
>
> Another direction is extending SoundCTM for tasks such as audio style transfer, bandwidth extension, or audio declipping, using pretrained diffusion models. For example, in the case of audio style transfer, a zero-shot music editing method like MusicMagus [16], which leverages cross-attention map guidance, could be adapted using SoundCTM's cross-attention mechanism. Similarly, sound enhancement methods that use loss-based guidance, such as CQT-Diff [17], could potentially benefit from a similar approach using SoundCTM.
>
> However, neither of these applications represents a trivial task that can be comprehensively addressed in a single paper. For example, in cross-attention control, challenges arise in managing the text embedding space and introducing appropriate constraints for guidance [16]. For sound enhancement, achieving data consistency between the observed sample and the enhanced sample remains a non-trivial challenge, even in LDM-based diffusion models.
>
> Addressing these non-trivial challenges and expanding the possibilities of SoundCTM through future research represents a meaningful direction for the audio research community.
>
> [14] Welker, S., Richter, J. and Gerkmann, T.. Speech enhancement with score-based generative models in the complex STFT domain. Interspeech 2022.
>
> [15] Serrà, J., Pascual, S., Pons, J., Araz, R.O. and Scaini, D., 2022. Universal speech enhancement with score-based diffusion. arXiv preprint arXiv:2206.03065.
>
> [16] Zhang, Y., Ikemiya, Y., Xia, G., Murata, N., Martínez-Ramírez, M.A., Liao, W.H., Mitsufuji, Y. and Dixon, S.. Musicmagus: Zero-shot text-to-music editing via diffusion models. IJICAI 2024.
>
> [17] Moliner, E., Lehtinen, J. and Välimäki, V.. Solving audio inverse problems with a diffusion model. ICASSSP 2023.

---

### Official Review · Reviewer_s79p · 2024-11-02

**Soundness:** 3
**Presentation:** 2
**Contribution:** 3
**Rating:** 6
**Confidence:** 2

**Summary:**

This paper proposes a new Sound Consistency Trajectory Model (SoundCTM) to accelerate the iterative trial-and-error process of sound design using audio generation models.

Compared with previous 1-step distillation methods, SoundCTM provides deterministic sampling capabilities, which enables acoustic consistency between 1-step and multi-step generations from the same initial noise latent.

Creators can use SoundCTM in the fast 1-step setting for exploratory work while benefitting from the higher quality of multi-step inference once a relevant generated sample has been chosen.

The authors claim the following contributions.
- Adaptation of the computer vision domain CTM training framework to the audio domain (called SoundCTM).
- A novel feature distance for distillation loss computation.
- v-sampling: a new cfg-like approach that consists in training both conditional and unconditional student models.
- SoundCTM-DiT-1B, a large scale distillation model that achieves sota 1-step and multi-step full-band text-to-sound generation on the AudioCaps evaluation dataset.

The results indicate better consistency between 1-step and multi-step generation compared with previous sota.

**Strengths:**

- Showcasing a potential downstream application of the proposed method is a nice addition to the work.
- Speed and inference consistency between 1-step/multi-step inference is a fundamental problem in the sound design space and this paper properly tackles this problem.

**Weaknesses:**

- It is not clear how v-sampling differs from CFG hence it is difficult to appreciate the relevance of the contribution.
- It is unclear why the proposed distilled model outperforms its teacher.

**Questions:**

- In v-sampling, it is not clear whether there is one or two student models involved. From the abstract it seems that there are two but not from section 3.3 and algorithm 2.
- In what sense is v-sampling different than CFG applied to the student model?
- What is the difference between the last row of Table 1 and the second to last row of Table 2? They do not indicate the same values although they seem to relate to the same experiment.
- How do you explain that SoundCTM-DiT-1B outperforms the teacher model?

---

> ### Author Response · Authors · 2024-11-22
>
> We extend our heartfelt thanks to the reviewer for their detailed evaluations and thoughtful comments. Below, we provide responses to each question and concern to clarify and strengthen our work.
>
> #### **Response to W1, W2, Q2, Q4**
> >**"W1.: It is not clear how v-sampling differs from CFG hence it is difficult to appreciate the relevance of the contribution."**
>
> $\nu$-sampling plays a role in consistency-trajectory "distillation" models akin to that of CFG in "diffusion" models. (Therefore, intuitively, as illustrated in Figure 13, it also helps alleviate ‘blurriness’ in the generated outputs.)
>
> Practically, as you can see in Figure 12 and Appendix D.4., $\nu$-sampling makes a significant contribution to improving the performance of multi-step sampling.
>
> >**"Q2: In what sense is v-sampling different than CFG applied to the student model?"**
>
> The key difference between CFG applied to student models and $\nu$ lies in the following:
> - When $\nu=1$, we choose a learned CFG-guided trajectory from the teacher model (the student models leaned various CFG-guided trajectory during distillation since we uniformly sample CFG from certain range.). In this case, inference trajectory is bounded by a chosen CFG-guided trajectory.
> - $\nu$ introduces additional degrees of freedom by not being bound by the learned CFG trajectory since we also use the unconditional trajectory.
>
> >**"- W2: It is unclear why the proposed distilled model outperforms its teacher."**
>
> >**"- Q4: How do you explain that SoundCTM-DiT-1B outperforms the teacher model?"**
>
> Regarding the ability to outperform the teacher model, our findings are limited to empirical results. Whether it can theoretically outperform the teacher model is left as future work.
>
> To further investigate why $\nu$-sampling achieves good performance during small jumps (multi-step), we provide the following considerations:
> - Small jumps on CFG-trajectory contains estimated error since the current DSM loss (Eq.(8)) learns conditional/unconditional scores rather than the "CFG-guided" score. (The accuracy of small jumps relies on the DSM loss).
> - The unconditional student model used in $\nu$-sampling compensates for these errors, functioning similarly to CFG-sampling in diffusion models.
> - The importance of the accuracy of unconditional student model is also empirically evident from the ablation study in Table 11. When the DSM loss is removed (i.e., when the small jumps of the unconditional student model are also inaccurate), we cannot see the benefits of $\nu$-sampling.
>
> We also provide additional explanations for potential concerns:
> - Do we still need to distill CFG-trajectory during distillation? => Yes, without this, even using $\nu$-sampling for 1-step generation, we cannot get good samples (see $\omega=1$ in Figure 12 (c).). This might highlight the important role of distilling the CFG-guided trajectory distillation, especially for 1-step generation.

---

> ### Author Response · Authors · 2024-11-22
>
> #### **Response to Q1**
> >**" Q1: In v-sampling, it is not clear whether there is one or two student models involved. From the abstract it seems that there are two but not from section 3.3 and algorithm 2."**
>
> In $\nu$-sampling, both the CFG-guided conditional student model $G_{\theta}(z_t, c_{text}, \omega, t, s)$ and the unconditional student model $G_{\theta}(z_t, \varnothing, \omega, t, s)$ are utilized. These two student models are trained simultaneously.
>
> #### **Response to Q3**
> >**"Q3:  What is the difference between the last row of Table 1 and the second to last row of Table 2? They do not indicate the same values although they seem to relate to the same experiment."**
>
> The difference lies in the checkpoints that we used. For a fair comparison, the results presented in Table 1 are all based on the same training iteration (18k iterations), while those in Table 2 correspond to 30k iterations.

---

### Official Review · Reviewer_exXk · 2024-11-05

**Soundness:** 2
**Presentation:** 2
**Contribution:** 2
**Rating:** 6
**Confidence:** 5

**Summary:**

The paper proposes Sound Consistency Trajectory Models (SoundCTM) for improving inference speed and sample quality in Text-to-Sound (T2S) tasks by combining score-based and consistency models. This work mainly adapting existing techniques from image generation for T2S.  From the reported results, SoundCTM can generate sound in few inference step.

**Strengths:**

1. The paper introduces Sound Consistency Trajectory Models (SoundCTM), an approach that unifies score-based and consistency models for Text-to-Sound (T2S) generation. This combination is interesting and reflects a novel direction within the T2S domain, especially in adapting techniques from image generation.

2. The paper evaluates SoundCTM at both 16kHz and 44.1kHz, showing an awareness of high-fidelity audio generation.

**Weaknesses:**

1. The paper proposes Sound Consistency Trajectory Models (SoundCTM) to improve inference speed. While the concept of combining score-based and consistency models for T2S generation is interesting, the paper does not convincingly establish the novelty of its contributions. Many of the methods, including ν-sampling and trajectory-based distillation, are either borrowed from existing frameworks or lack adequate differentiation from prior work. A clearer and more distinct innovation beyond adapting methods from image-based models is needed to justify the novelty of this work.

2. The paper emphasizes deterministic sampling as a unique advantage of SoundCTM, particularly for preserving semantic content across sampling steps. However, this advantage remains largely theoretical within the paper, with insufficient empirical evidence demonstrating the effectiveness of deterministic sampling in real production scenarios. The provided spectrograms and subjective evaluations do not strongly support the claimed benefits, and thus, the deterministic aspect feels overstated relative to its demonstrated impact.

3. The paper highlights full-band sound generation. However, all previous work can support full-band sound generation by simply replacing 16kHz audio data with 44.1kHz data.

4. In terms of performance, SoundCTM shows some advantages over AudioLCM and ConsistencyTTA. However, as SoundCTM scales to a 1B parameter model, it is difficult to determine whether the improvements are due to model scaling alone.

5. From an architectural perspective, the proposed DiT framework closely follows the architecture of Stable Audio 2.0.

6. In the related work section, the paper claims that "when developing sound generative models suitable for production use, LDM-based models present a better choice as the base model." It is difficult to agree with this claim. Previous works, such as AudioGen, have also demonstrated good performance in T2S. For example, DiffSound shows that using a discrete diffusion model yields better performance than an AR-based baseline. After that, AudioGen demonstrates that AR-based models can also achieve good performance with different configurations. Given the rapid development of T2S, it is premature to make such a claim. In the field of image generation, some studies have shown that AR-based models can outperform LDM-based models. Additionally, discrete diffusion models remain an important research direction.

**Questions:**

Please refer to Weaknesses.

1. What your training dataset?

---

> ### Author Response · Authors · 2024-11-22
>
> We would like to express our gratitude to the reviewer for their providing constructive feedback. In the following, we address each question and concern with the aim of clarifying and enhancing our work.
>
> #### **Response to W1**
> >**"The paper proposes Sound Consistency Trajectory Models (SoundCTM) to improve inference speed. While the concept of combining score-based and consistency models for T2S generation is interesting, the paper does not convincingly establish the novelty of its contributions. Many of the methods, including ν-sampling and trajectory-based distillation, are either borrowed from existing frameworks or lack adequate differentiation from prior work. A clearer and more distinct innovation beyond adapting methods from image-based models is needed to justify the novelty of this work."**
>
> We acknowledge the reviewer's observation that our proposed techniques such as trajectory-based distillation and $\nu$-sampling, are indeed inspired by the ideas developed in the computer vision domain. However, we respectfully ask the reviewer to reconsider whether the novelty of a research work should be evaluated solely based on technical or methodological extensions [1].
>
> First and foremost, let us clarify what we aimed to achieve in this work and outline our contributions.
>
> **[What we want to achieve in this paper]**
>
> The primary aim of this paper is not merely to accelerate inference speed. Rather, we aim to develop a model/framework that addresses a critical gap between existing sound generative models and their real-world applications: the ability to flexibly adapt to user demands in sound creation workflows. Specifically, SoundCTM provides a unique capability, "deterministic sampling via $\gamma$-sampling", not achieved by any prior sound generative model—enabling an efficient trial-and-error process with 1-step generation and, once users identify the desired conditions and hyperparameters through this process, they can obtain higher sample quality via deterministic multi-step sampling while preserving semantic content (We statistically confirm this capability in the next response.).
>
> **[Our contribution]**
>
> For developing the model having such capability of filling the gap that we mentioned, we focus on the feature of the sampling flexibility of consistency trajectory distillation among many existing distillation models. We think that, to achieve our goal, highlighting its feature itself constitutes one of novel contributions to the sound community [1]. Moreover, instead of validating the our approach under minimal experimental setups, we demonstrate its feasibility at 44.1 kHz, which is a standard requirement in real-world sound applications. Our contributions also include the proposed submodules (we conducted extensive additional experiments and ablation studies in Appendix D.), the established training recipes, and the pretrained checkpoints—all of which are planned to be open-sourced upon acceptance. We believe these contributions collectively offer a significant and novel contributions for advancing further research in the sound community.
>
> **[Our sincere request]**
>
> As we mentioned at the beginning of this response, we sincerely ask the reviewer to re-evaluate our contributions from a fair and balanced standpoint along with the subsequent responses.
>
> [1] https://medium.com/@black_51980/novelty-in-science-8f1fd1a0a143

---

> > ### Comment · Reviewer_exXk · 2024-11-22
> >
> > Thank you for your detailed response:
> >        Please respect each reviewer's view, especially for the novelty. Different researchers may have different views about the novelty contribution. You cite a blog to ask the reviewer to learn to review is disrespectful. Novelty, just one aspect to give 3 scores in the previous stage. Many reviewers also raise concerns about the semantic content preservation and performance comparison between AudioLCM. This is why we need a rebuttal stage to solve these concerns.
> >
> > Considering the authors claim to open-source the models. I will increase my score to 5.

---

> > > ### Author Response · Authors · 2024-11-25
> > >
> > > Dear Reviewer exXk,
> > >
> > > We sincerely appreciate the time and effort you dedicated to carefully reviewing our responses.
> > > We deeply regret and apologize for any inappropriate expressions in our replies.
> > >
> > > Furthermore, we are truly grateful for your kind agreement to update the rating.
> > > If it is not too much trouble, we would greatly appreciate it if you could also update the 'Rating' in the console to the score you mentioned.
> > >
> > > Best regards,
> > >
> > > Authors

---

> ### Author Response · Authors · 2024-11-22
>
> #### **Response to W2**
> >**"W2. The paper emphasizes deterministic sampling as a unique advantage of SoundCTM, particularly for preserving semantic content across sampling steps. However, this advantage remains largely theoretical within the paper, with insufficient empirical evidence demonstrating the effectiveness of deterministic sampling in real production scenarios. The provided spectrograms and subjective evaluations do not strongly support the claimed benefits, and thus, the deterministic aspect feels overstated relative to its demonstrated impact."**
>
> We agree that the demonstration of the deterministic capability was not sufficient, as it lacked an objective evaluation to verify this aspect. To further evaluate SoundCTM's semantic content preservation in deterministic sampling, we calculated sample-wise reconstruction metrics. Specifically, we computed LPAPS using the CLAP audio encoder (CLAP-LPAPS) [2] via the toolkit [3] with 957 samples from the AudioCaps test set.
>
> **We computed the CLAP-LPAPS between samples generated by SoundCTM-DiT-1B with 1-step and 16-step across several values of $\gamma$.**
>
> Furthermore, to benchmark these CLAP-LPAPS values themselves, we also calculated scores for:
> - **"VAE-reconstruction"**: Ground-truth audio samples from the AudioCaps test set and their reconstructed samples using the VAE in SoundCTM-DiT-1B. **This indicates a signal-level reconstruction score**.
> - **"AC GT test samples"**: Ground-truth audio samples from the AudioCaps test set and SoundCTM-DiT-1B's 1-step generated samples. **This roughly indicates a semantic-level random score within the same distribution.**
>
> We report the average and standard deviation of the CLAP-LPAPS scores from 957 samples.
>
> |Method|$\gamma$|CLAP-LPAPS$\downarrow$|
> |----|----|----|
> |VAE-reconstruction|-|2.548 ± 0.394|
> |**Deterministic**|0| **3.343 ± 0.737**|
> |Stochastic|0.2|5.209 ± 0.565|
> ||0.5|5.212 ± 0.557|
> ||1.0 (CM-sampling)|5.341 ± 0.651|
> |AC GT test samples|-|6.073 ± 0.411|
>
> Based on these results, we believe that the deterministic sampling of SoundCTM demonstrates the capability of preserving semantic content from a statistical perspective.
>
> [2] Manor, H. and Michaeli, T.. Zero-shot unsupervised and text-based audio editing using DDPM inversion. ICML 2024.
>
> [3] https://github.com/HilaManor/AudioEditingCode/tree/codeclean/evals

---

> ### Author Response · Authors · 2024-11-22
>
> #### **Response to W3**
> >**"W3. The paper highlights full-band sound generation. However, all previous work can support full-band sound generation by simply replacing 16kHz audio data with 44.1kHz data."**
>
> In sound research field, enabling full-band audio generation is far from trivial. It cannot be achieved simply by replacing the data.
> - For example, in the field of neural audio compression models or vocoders used in the pre- and post-components of generative models, research aimed at achieving high-fidelity performance is still ongoing [4,5].
> - Similarly, in generative models, concerns have been raised that merely adapting hyperparameters to accommodate differences in resolution, without modifying the model architecture for example, is insufficient (e.g., Limitations noted in [6]). Performance degradation has also been observed in practice (e.g., AudioLDM2-16kHz and AudioLDM2-48kHz in Tables 1, 2, and 4 of the literature [7]).
>
> Additionally, modifying those certain components often necessitates adjustments or retraining of other components, leading to increase of non-trivial training costs.
>
> Considering of these situations, as also mentioned in "Response to W1", open-sourcing both the training recipe and the checkpoints of our full-band sound generation models also represents a meaningful contribution to this community.
>
> [4] Zhu, G., Caceres, J.P., Duan, Z. and Bryan, N.J., 2024. MusicHiFi: Fast high-fidelity stereo vocoding. IEEE SLT.
>
> [5] Kumar, R., Seetharaman, P., Luebs, A., Kumar, I. and Kumar, K.. High-fidelity audio compression with improved rvqgan. NeurIPS 2023.
>
> [6] Kreuk, F., Synnaeve, G., Polyak, A., Singer, U., Défossez, A., Copet, J., Parikh, D., Taigman, Y. and Adi, Y.. Audiogen: Textually guided audio generation. ICLR 2023.
>
> [7] Evans, Z., Carr, C.J., Taylor, J., Hawley, S.H. and Pons, J.. Fast timing-conditioned latent audio diffusion. ICML 2024.

---

> ### Author Response · Authors · 2024-11-22
>
> #### **Response to W4**
> >**"W4. In terms of performance, SoundCTM shows some advantages over AudioLCM and ConsistencyTTA. However, as SoundCTM scales to a 1B parameter model, it is difficult to determine whether the improvements are due to model scaling alone."**
>
> Since SoundCTM demonstrates its advantages particularly in 1-step generation against both ConsistencyTTA and SoundCTM from objective evaluation, we focus on comparisons from this perspective. As discussed in detail below, we believe that **model scaling alone is not the sole factor** contributing to the performance in either comparison.
>
> **[ConsistencyTTA vs. SoundCTM]**
>
> We think the performance difference between these two models primarily stems from their consistency matching methods, with a secondary factor being the difference in model size (557M UNet in ConsistencyTTA vs. 866M UNet in SoundCTM). Specifically, ConsistencyTTA distills the teacher solver using 1-step sampling (local consistency matching, as shown in Figure 10(a) in [8]), whereas SoundCTM employs multi-step sampling (soft consistency matching, also illustrated in Figure 10(a) in [8]).
> Figure 10 (a) in [8] reports that using soft consistency matching results in much better 1-step generation performance.
>
> **[AudioLCM vs. SoundCTM]**
>
> The performance difference between AudioLCM and SoundCTM in 1-step generation is challenging to pinpoint due to various factors, including differences in VAE and vocoder architecture, the student/teacher model architecture, the teacher model's training dataset, and the exponential moving average values used during distillation ($\mu$=0.95 in contrast to $\mu=0.999$ in ConsistencyTTA/SoundCTM).
>
> However, it seems unlikely that model scaling or consistency matching methods are the primary reasons for the performance gap, according to some experimental results in AudioLCM paper [9].
> - Why seems not from model scaling?: Figure 2(b) in [9] shows a significant FAD gap (about 2.4 points) between 1-step performance (FAD around 4) and multi-step performance (FAD around 1.6). Table 7 in [9] further indicates that large-scale training (up to 2.4B parameters) only results in a minor improvement in FAD (maximum 0.17 points). Given this observation, we do not believe that scaling up AudioLCM to 866M parameters would enable it to reach the 1-step performance of SoundCTM (866M paramters in this setup).
> - Why seems not consistency matching?. AudioLCM also uses soft consistency matching (20 steps with a DDIM solver), which aligns with SoundCTM.
>
> **[Uniqueness of SoundCTM]**
>
> While we have discussed the potential causes of the performance differences in 1-step generation, it is worth emphasizing that SoundCTM offers a unique flexible sampling capability (as detailed in "Response to W2" we conduct additional objective evaluations).
> As mentioned in "Response to W1", this unique feature allows users to perform efficient trial-and-error with 1-step generation and then, once the desired conditions are identified, utilize multi-step deterministic sampling to produce higher-quality samples while preserving semantic content.
>
> [8] Kim, D., Lai, C.H., Liao, W.H., Murata, N., Takida, Y., Uesaka, T., He, Y., Mitsufuji, Y. and Ermon, S.. Consistency trajectory models: Learning probability flow ode trajectory of diffusion. ICLR 2024.
>
> [9] Liu, H., Huang, R., Liu, Y., Cao, H., Wang, J., Cheng, X., Zheng, S. and Zhao, Z., 2024. AudioLCM: Text-to-Audio Generation with Latent Consistency Models. arXiv preprint arXiv:2406.00356.
>
> #### **Response to W5**
> >**W5. From an architectural perspective, the proposed DiT framework closely follows the architecture of Stable Audio 2.0.**
>
> We agree that the architecture is indeed close to Stable Audio 2.0. However, proposing a teacher model architecture is beyond the scope of this paper. Our focus is on developing a new distillation model, rather than improving the architecture of pretrained models.
>
> #### **Response to Q1**
> >Q1. What your training dataset?
>
> We used AudioCaps dataset

---

> ### Author Response · Authors · 2024-11-22
>
> #### **Response to W6**
> >**"W6. In the related work section, the paper claims that "when developing sound generative models suitable for production use, LDM-based models present a better choice as the base model." It is difficult to agree with this claim. Previous works, such as AudioGen, have also demonstrated good performance in T2S. For example, DiffSound shows that using a discrete diffusion model yields better performance than an AR-based baseline. After that, AudioGen demonstrates that AR-based models can also achieve good performance with different configurations. Given the rapid development of T2S, it is premature to make such a claim. In the field of image generation, some studies have shown that AR-based models can outperform LDM-based models. Additionally, discrete diffusion models remain an important research direction."**
>
> We appreciate your fair and insightful observations. As you pointed out, AR models, non-AR models, LDM-based models, and discrete diffusion models are all competing with each other in this domain. Your comments have been incorporated into the manuscript.

---

> ### Author Response · Authors · 2024-11-22
>
> Once again, we respectfully request you to re-evaluate our work from a fair and balanced standpoint. Through the series of our responses, we think we have addressed all the concerns raised.
>
> To briefly recap:
> - Our primary goal was to bridge the gap between existing sound generative models and real-world applications by proposing a new distillation model.
> - As one of unique contributions to the sound research field, we presented a model that enables creators to perform efficient trial-and-error with 1-step, while also generating higher-quality samples with preserveing semantic content-a capability that has not been achieved by prior sound generative models.
> - We confirmed the effectiveness of this distinctive feature, particularly the deterministic sampling capability, through an additonal objective evaluation from a statistical perspective.
> - Given that achieving high-fidelity sound generation is non-trivial in the sound research field, we experimentally demonstrated that our proposed model is feasible for full-band sound generation.
> - Finally, we plan to release the training and inference codebase, along with model checkpoints, upon acceptance.
>
> We firmly believe these contributions collectively represent a significant and novel advancement, paving the way for further research in this community.

---

### Official Review · Reviewer_FAko · 2024-11-05

**Soundness:** 3
**Presentation:** 3
**Contribution:** 3
**Rating:** 8
**Confidence:** 5

**Summary:**

This paper presents a diffusion-based text-to-audio generation system within the Consistency Trajectory Model (CTM) framework, named SoundCTM. SoundCTM is developed using distillation loss with a teacher model, and incorporates a novel sampling algorithm that enables Classifier-Free Guidance (CFG) integration with the student model. Experimental results demonstrate that SoundCTM achieves state-of-the-art performance in 1-step and few-step generation for 44.1 kHz full-band samples, offering a practical tradeoff between generation quality and inference speed for real-world applications.

**Strengths:**

1. Results indicate that the proposed large-scale SoundCTM-Dit-1B achieves state-of-the-art performance in text-to-audio generation in the full-band setting (mono at 44.1 kHz).
2. Instead of using the L2-norm loss between target and output values within the latent domain, this paper introduces a network trained with CTM loss, leveraging a teacher network for feature extraction and measurement. Results show that the proposed teacher loss achieves superior performance.
3. This paper integrates Classifier-Free Guidance (CFG) within the CTM framework by distilling the CFG-related PF ODE trajectory during training and using it as the CFG condition. Additionally, a new sampling strategy has been developed to enable CFG in 1-step generation.

**Weaknesses:**

1. The comparison results show that the proposed system achieves state-of-the-art performance in single-step sampling, while SoundCTM does not surpass baseline models (e.g., AudioLCM, also a diffusion model with CTM) when increasing the sampling steps (from 2 steps onward). Additionally, there is a lack of comparison in inference speed between different distillation models.
2. Further explanation is required for why baseline models outperform SoundCTM in few-step sampling. Moreover, the CLAP score is the only metric related to semantic alignment (text), yet it does not seem well-aligned with other metrics, such as FAD and IS. It would therefore be beneficial to conduct human evaluations for each experiment to better illustrate performance.
3. The authors claim that SoundCTM-Dit-1B achieves the best results in full-band generation tasks. However, SoundCTM used for comparisons among 16 kHz models employs a UNet-based backbone, resulting in unaligned experiments. The paper lacks sufficient explanations, ablation studies, or comparisons to demonstrate the effectiveness of each submodule. The authors could address this by adding SoundCTM-DIT medium or small models for 16 kHz comparisons and SoundCTM-UNet-1B for full-band comparisons.

Overall, this is an interesting paper exploring the potential of consistency models in sound generation. The model is well-structured, but some modules require further explanation, and additional experiments are needed to illustrate the contributions of each section. I am open to adjusting the score if the authors can address these requirements.

**Questions:**

1. Is the hyperparameter 𝑣 mentioned in Section 3, which is used for sampling, trainable? If not, how did the authors determine its value? The same question applies to the scale of CFG 𝑤
2. For the results in Table 2, what is the best performance SoundCTM can achieve? With additional sampling steps, can SoundCTM surpass the quality of AudioDLM2-large?
3. What are the differences between SoundCTM and AudioLCM, aside from the teacher CTM loss proposed for training? Both systems are diffusion-based generation models employing consistency strategies.
4. The authors also state that SoundCTM and AudioLCM show clear trade-offs in multi-step generation. What is the inference efficiency of these two systems? The authors mentioned early on that the proposed system is intended to overcome the barrier of slow inference speeds encountered during trial-and-error.

---

> ### Author Response · Authors · 2024-11-21
>
> We sincerely appreciate the time and effort the reviewer has devoted to providing detailed and thoughtful feedback. In the following response, we would like to address the reviewer' concerns regarding of our in a clear and comprehensive manner.
>
> #### **Response to W1**
> >**"W1. The comparison results show that the proposed system achieves state-of-the-art performance in single-step sampling, while SoundCTM does not surpass baseline models (e.g., AudioLCM, also a diffusion model with CTM) when increasing the sampling steps (from 2 steps onward). Additionally, there is a lack of comparison in inference speed between different distillation models. Further explanation is required for why baseline models outperform SoundCTM in few-step sampling. "**
>
> - We measured the inference speeds on an NVIDIA A100 GPU with a batch size of one and reported in Table 8 in the revised paper.
>
> - The performance differences between SoundCTM and AudioLCM, as shown in Table 2 and Table 9, which we will cover at "Response to W2" and Appendix D.3., are multifaceted, making it non-trivial to identify a definitive cause.
> There are differences in the architectures of both the student and teacher models—not only in the generator but also in other components such as the VAE and vocoder—as well as in the training datasets used for the teacher model. Moreover, based on the additional human evaluation in Table 9, SoundCTM demonstrates better performance.
>
> #### **Response to Q3**
> >**"Q3. What are the differences between SoundCTM and AudioLCM, aside from the teacher CTM loss proposed for training? Both systems are diffusion-based generation models employing consistency strategies."**
>
> We would like to highlight once again that while AudioLCM/ConsistencyTTA are trained using Consistency Distillation (CD; anytime-to-zero jump) [1], SoundCTM employs Consistency Trajectory Distillation (CTD; anytime-to-anytime jump) [2]. This fundamental difference results in distinct characteristics between the two models especially for sampling.
>
> **A key difference is that AudioLCM/ConsistencyTTA only supports stochastic sampling, whereas SoundCTM can control the level of stochasticity via $\gamma$-sampling.** The deterministic sampling is a particularly significant feature that sets SoundCTM apart from other two distillation models in practical use. It enables efficient trial-and-error processes (such as exploring the desired samples based on text conditioning, random seeds, and hyperparameters) using 1-step generation. Subsequently, with the same conditions, deterministic multi-step sampling can be employed to obtain higher-quality samples while preserving the semantic content. **In contrast, with stochastic sampling in AudioLCM/ConsistencyTTA, the semantic meanings change depending on the number of sampling steps.** (Please also see Figures 3, 6, and Table 7 in the paper.)
> This makes it difficult for users to predict or control the outputs, adding unnecessary effort to real-world workflows.
>
> #### **Response to Q4**
> >**" Q4. The authors also state that SoundCTM and AudioLCM show clear trade-offs in multi-step generation. What is the inference efficiency of these two systems? The authors mentioned early on that the proposed system is intended to overcome the barrier of slow inference speeds encountered during trial-and-error."**
>
> From the perspective of trial-and-error processes, in Table 8, the difference in inference efficiency between AudioLCM and SoundCTM is that SoundCTM allows trial-and-error using only 1 step (e.g., 0.141 [sec.]), whereas AudioLCM requires multiple steps, which is required to achieve the desired sample quality (e.g., 0.220 [sec.] for 4 steps) due to its inherent stochastic sampling. Of course, the practical inference speed varies depending on factors such as implementation and GPU type, but in terms of the number of sampling steps, SoundCTM is more efficient than AudioLCM in this scenario.
>
> [1] Song, Y., Dhariwal, P., Chen, M. and Sutskever, I., 2023. Consistency models. ICML 2023.
>
> [2] Kim, D., Lai, C.H., Liao, W.H., Murata, N., Takida, Y., Uesaka, T., He, Y., Mitsufuji, Y. and Ermon, S., 2023. Consistency trajectory models: Learning probability flow ode trajectory of diffusion. ICLR 2024.

---

> ### Author Response · Authors · 2024-11-21
>
> #### **Response to W2**
>
> >**"W2. Moreover, the CLAP score is the only metric related to semantic alignment (text), yet it does not seem well-aligned with other metrics, such as FAD and IS. It would therefore be beneficial to conduct human evaluations for each experiment to better illustrate performance."**
>
> Since we agree with the reviewer's concern, we conducted additional subjective evaluation for the 16 kHz setting. Due to the limited time during the rebuttal period, we focused on comparing 16 kHz models of AudioLCM and SoundCTM, as this seems to be a particular concern for the reviewer. Please find the experimental detail sin Appendix D.3.
> - Results: In contrast the trends observed in FAD_vgg in Table 2, SoundCTM outperformed AudioLCM on this human evaluation, even when comparing the results at 4 steps.
> - Reason: One possible reason for this is the artifacts in some samples generated by AudioLCM, as observed when listening to its outputs. These artifacts may have influenced the results of the human evaluation.
> - We share examples of the generated samples from AudioLCM and SoundCTM on the following link: https://drive.google.com/drive/folders/1SAFwzhQ5KlSX17aoVTvNzDhDrmr6W99l?usp=sharing

---

> ### Author Response · Authors · 2024-11-21
>
> #### **Response to W3**
> >**"W3. The authors claim that SoundCTM-Dit-1B achieves the best results in full-band generation tasks. However, SoundCTM used for comparisons among 16 kHz models employs a UNet-based backbone, resulting in unaligned experiments. The paper lacks sufficient explanations, ablation studies, or comparisons to demonstrate the effectiveness of each submodule. The authors could address this by adding SoundCTM-DIT medium or small models for 16 kHz comparisons and SoundCTM-UNet-1B for full-band comparisons."**
>
> We conducted additional ablation studies for each submodule, and the results are presented in Appendix D.4. These results will demonstrate each submodule influences the performance of SoundCTM-DiT-1B.
>
> **The overview of the additional ablation studies are as follows**:
> 1. A performance comparison between SoundCTM-UNet-1B and SoundCTM-DiT-1B at 44.1 kHz.
> 2. Ablation studies on submodules of SoundCTM-DiT-1B at 44.1kHz:
>    - Loss function: Comparison of CTM+DSM loss vs. CTM loss.
>    - Feature extractor on DiT: w/. vs. w/o. teacher's feature extractor
>
> All evaluations were conducted on the AudioCaps test set, consisting of 957 samples. Please find in detail in Appendix D.4.
>
> - **[Comparison between SoundCTM-UNet-1B and SoundCTM-DiT-1B on 44.1kHz]**
> We reported the results in Table 10. The result indicates that SoundCTM-DiT-1B outperforms SoundCTM-UNet-1B, particularly 1-step generation.
>
> - **[Ablation study on loss function of DiT on 44.1kHz]**
> We provide a performance comparison of SoundCTM-DiT-1B with and without the DSM loss in Table 11. Note that the DSM loss (Eq.(8)) serves to improve the accuracy of approximating the small jumps during the training.
> The result shows that while the performance is comparable for 1-step sampling, incorporating the DSM loss leads to improved performance as the number of the sampling steps increases.
>
> - **[Ablation study on teacher's feature extractor of DiT on 44.1kHz]**
> We report the results of the ablation study on the teacher's feature extractor in SoundCTM-DiT-1B in Table 12. The results indicate that the performance for 1-step generation are comparable. However, as the number of sampling steps increased, using feature extractor demonstrates better performance. On the other hand, in SoundCTM at the 16 kHz setting (see also Table 13), employing the teacher's feature extractor consistently outperforms the case where the teacher's feature extractor is not used, even for 1-step sampling.
> There are notable differences between the 16 kHz UNet model and the 44.1 kHz DiT model, including the VAE part, using vocoder or not, and the sampling frequency. To clarify the differences in trends for 1-step generation, further ablation studies are needed.
> However, these results demonstrate that employing the teacher's feature extractor leads to improved performance in multi-step generation.
>
> To sum up, based on all the ablation studies, we believe that each submodule contributes to the overall performance improvement of SoundCTM-DiT-1B at 44.1 kHz for both 1-step and multi-step sampling.

---

> ### Author Response · Authors · 2024-11-22
>
> #### **Response to Q1**
> >**"Q1. Is the hyperparameter 𝑣 mentioned in Section 3, which is used for sampling, trainable? If not, how did the authors determine its value? The same question applies to the scale of CFG $\omega$"**
>
> Neither $\omega$ nor $\nu$ are trainable parameters; they are hyperparameters set during inference. For $\omega$, we selected the value that generate good samples in the teacher model. For $\nu$ we set it to $\nu=1$ for fewer steps and increased its value for a larger number of steps. Further evaluation for 1-step and 16-step generation based on different $\omega$ and $\nu$ is reported in Appendix D.4 and in Table 12.
>
> #### **Response to Q2**
> >**"Q2. For the results in Table 2, what is the best performance SoundCTM can achieve? With additional sampling steps, can SoundCTM surpass the quality of AudioDLM2-large?"**
>
> Yes. SoundCTM can achieve comparable performance with AudioLDM2-AC-large.
> Below, we present a comparison between the results of SoundCTM with up to 16 steps and AudioLDM2-AC-Large on 16kHz setup. Since the checkpoint for AudioLDM2-AC-Large is not publicly available, we were unable to compare metrics such as IS_passt, CLAP score, or human evaluations. However, it demonstrates comparable performance in terms of the FAD_vgg.
>
> |Models|# of sampling steps|$\omega$|$\nu$|FAD_vgg $\downarrow$|KL_passt $\downarrow$|IS_passt $\uparrow$|CLAP $\uparrow$|
> |----|----|----|---|---|----|----|---|
> |AudioLDM 2-AC-Large|200|3.5|-|1.42|**0.98**|-|-|
> |SoundCTM|8|3.0|1.5|1.45|1.20|7.98|0.46|
> ||16|3.0|2.0|**1.38**|1.19|8.24|0.46|

---

> ### Comment · Reviewer_FAko · 2024-11-24
>
> Thanks for all the replies from the author. Most of the questions are included and I am happy to adjust the score based on the responses.

---

### Author Response · Authors · 2024-12-03

Dear Reviewers and ACs,

As the discussion period comes to an end, we would like to express our sincere gratitude to all the reviewers for their constructive feedback and to the ACs for their efforts in facilitating the discussion period.

At this point, we would like to particularly thank all the reviewers for dedicating their time and effort to providing valuable feedback throughout both the first-round review and the rebuttal period.

#### **[At the first-round review]**

We were grateful for the recognition of SoundCTM as **a novel and interesting direction/approach** for text-to-sound generative models (FAko, exXk, kUmE) and a **well-structured model** (FAko) that leverages **our proposed feature extractor and $\nu$-sampling** (FAko, kUmE).
Moreover, we appreciated the reviewers' acknowledgment of our model’s ability to **generate high-quality, high-fidelity sound** at 44.1 kHz (FAko, exXk), as well as their recognition of the **flexible sampling that properly addresses a fundamental problem in the sound design space** (s79p, kUmE).
We were also encouraged by the **positive remarks regarding the potential downstream applications** of our model (s79p).

#### **[During the rebuttal period]**

We endeavored to address nearly all concerns raised by all the reviewers through new additional experiments/ablation studies and discussions. For instance, we demonstrated **each proposed submodule contributes to the overall performance** (FAko), showed that the **deterministic sampling is also effective in objective evaluations** (exXk), clarified the **intuitive explanation and effectiveness of ν-sampling** (FAko, s79p, kUmE), and explored **further potential downstream applications** (kUmE).
Furthermore, we elaborated on the **distinctions between SoundCTM and other models** (e.g., distillation-based text-to-sound generative models, as well as flow-based models) both from fundamental and practical perspectives (FAko, exXk, kUmE).

In conclusion, we would like to once again extend our heartfelt gratitude to all the reviewers for their valuable review of not only our initial submission but also our responses during the rebuttal period. We deeply appreciate FAko, exXk, and kUmE for providing additional responses to our rebuttal.

Warm regards,

Authors

---

### Meta-Review · Area_Chair_FYx1 · 2024-12-21

**Metareview:**

This paper proposes Sound Consistency Trajectory Models (SoundCTM), a text-to-sound generation framework that unifies score-based diffusion and consistency modeling. The core objective is to enable both fast trial-and-error via high-quality one-step sampling and refined multi-step sampling at full-band audio (44.1 kHz).

All four reviewers provided positive ratings, with final scores of 8, 8, 6, and 6. SoundCTM offers a valuable step forward in text-to-sound research by combining quick inference for iterative design with deterministic multi-step sampling for higher quality. The authors’ rebuttal addressed concerns about method, experiments, and contributions. While some novelty questions remain around adopting consistency trajectory models from vision, the overall consensus is that domain-specific engineering, large-scale experiments, and open-sourced code represent a substantial contribution to the T2S field.

Hence, I recommend acceptance. The authors should incorporate the additional results and clarifications in rebuttal to the revised version.

**Additional Comments On Reviewer Discussion:**

During the review process, Reviewer FAko highlighted the potential for exploring consistency models in the context of sound generation, while also requesting more comprehensive explanations of individual modules and further experimental results to clarify the function of each component. Reviewer s79p, maintaining a rating of 6, requested further clarifications of the proposed approach. Reviewer kUmE, impressed by SoundCTM's practical value in enabling deterministic sampling for real-world sound design applications and its capabilities in zero-shot bandwidth extension, increased their rating to 8 following the rebuttal phase. Reviewer exXk initially expressed reservations regarding the novelty of the work. Although the rebuttal addressed most reviewer concerns, Reviewer exXk initially increased their score only to 5. However, after further discussion, the reviewer acknowledged the value of adapting methods from other domains to audio generation and ultimately raised the score to 6.
Given the final positive ratings, I recommend accepting this paper.

---

### Decision · Program_Chairs · 2025-01-22

Accept (Poster)